# A surface proton antenna in carbonic anhydrase II supports lactate transport in cancer cells

Sina Ibne Noor[1], Somayeh Jamali[1], Samantha Ames[1], Silke Langer[1], Joachim W Deitmer[1], Holger M Becker[1,2]*

[1]Division of General Zoology, Department of Biology, University of Kaiserslautern, Kaiserslautern, Germany; [2]Institute of Physiological Chemistry, University of Veterinary Medicine Hannover, Hannover, Germany

**Abstract** Many tumor cells produce vast amounts of lactate and acid, which have to be removed from the cell to prevent intracellular lactacidosis and suffocation of metabolism. In the present study, we show that proton-driven lactate flux is enhanced by the intracellular carbonic anhydrase CAII, which is colocalized with the monocarboxylate transporter MCT1 in MCF-7 breast cancer cells. Co-expression of MCTs with various CAII mutants in *Xenopus* oocytes demonstrated that CAII facilitates MCT transport activity in a process involving CAII-Glu69 and CAII-Asp72, which could function as surface proton antennae for the enzyme. CAII-Glu69 and CAII-Asp72 seem to mediate proton transfer between enzyme and transporter, but CAII-His64, the central residue of the enzyme's intramolecular proton shuttle, is not involved in proton shuttling between the two proteins. Instead, this residue mediates binding between MCT and CAII. Taken together, the results suggest that CAII features a moiety that exclusively mediates proton exchange with the MCT to facilitate transport activity.

DOI: https://doi.org/10.7554/eLife.35176.001

*For correspondence:
Holger.Becker@tiho-hannover.de

Competing interests: The authors declare that no competing interests exist.

## Introduction

Many tumor cells, especially those that reside in a hypoxic environment, rely on glycolysis to meet their increased demand for energy and biosynthetic precursors (*Hanahan and Weinberg, 2011*; *Schulze and Harris, 2012*; *Parks et al., 2013*). As a consequence, they produce considerable amounts of acid and lactate, which have to be removed from the cell so as to prevent intracellular lactacidosis and suffocation of their metabolism. Lactate efflux from cancer cells is mediated primarily by the monocarboxylate transporters MCT1 (*SLC16A1*) and MCT4 (*SLC16A3*), both of which transport lactate together with a proton across the cell membrane (*Poole and Halestrap, 1993*; *Parks et al., 2013*). This MCT-mediated H+ efflux can exacerbate extracellular acidification and thus support the formation of a hostile environment, which favors tumor growth and allows tumor cells to escape conventional cancer therapies (*Kennedy and Dewhirst, 2010*; *Parks et al., 2011*, *2013*; *Gillies et al., 2012*). Expression of MCT1 is not altered with varying oxygen tension, whereas expression of MCT4 has been shown to be upregulated in different cancer cells under hypoxia (*Ullah et al., 2006*). However, expression of the two isoforms strongly varies among different tumor species and cell lines. In the MCF-7 breast cancer cell line, used in this study, lactate flux is exclusively mediated by MCT1 under both normoxic and hypoxic conditions (*Jamali et al., 2015*).

Another family of key proteins in tumor acid/base regulation is the carbonic anhydrases, which catalyze the reversible hydration of $CO_2$ to $HCO_3^- + H^+$. The role of the cancer-specific isoform CAIX in tumor development and progression has been studied extensively, but physiological data on CAII in cancer cells are still relatively scarce. CAII has been found in different types of brain

tumors, with the most malignant species exhibiting the strongest expression. In addition, this isoform has been linked to poor prognosis in patients suffering from those tumor types (*Parkkila et al., 1995*; *Haapasalo et al., 2007*). Furthermore, CAII has been suggested to mediate the malignant behavior of pulmonary neuroendocrine tumors (*Zhou et al., 2015*). Experiments on breast cancer cells demonstrated that CAII is upregulated in the highly tumorigenic MDA-MB-231 cell line when the cells are exposed to the chemotherapeutic drug doxorubicin (*Mallory et al., 2005*). On the other hand, a reduction in CAII expression was proposed to promote tumor cell motility and to contribute to tumor growth and metastasis in non-small cell lung cancer and gastric carcinoma (*Chiang et al., 2002*; *Li X-J et al., 2012*). CAII expression was also found to be associated with tumor differentiation and poor prognosis in patients with pancreatic cancer (*Sheng et al., 2013*), suggesting a differential function of CAII in different tumor types.

Catalysis by CA involves two distinct and separate stages: the interconversion of $CO_2$ and $HCO_3^-$ followed by the transfer of an $H^+$ to the bulk solution to regenerate the zinc-bound hydroxide, the latter being the rate-limiting step in the overall reaction (*Tu et al., 1989*; *Lindskog, 1997*). In CAII, the fastest of the α-CAs (*Steiner et al., 1975*; *Boone et al., 2014*), $H^+$ transfer between the zinc-bound water and the solvent surrounding the enzyme is facilitated by the side chain of His64, which shuttles $H^+$ between the bulk solvent and a network of well-ordered hydrogen-bonded water molecules in the enzyme's active-site cavity (*Fisher et al., 2007a*). Furthermore, the $H^+$ transfer efficiency of His64 is fine-tuned by several hydrophilic residues (Tyr7, Asn62 and Asn67), which contribute to the stabilization of the intervening water molecules within the active-site cavity and which influence the orientation and the $pK_a$ value of His64 (*Fisher et al., 2007b*). These activities mean that the His64 residue is essential for CAII to reach full enzymatic activity. On the basis of work using an algorithm for mapping proton wires in proteins, *Shinobu and Agmon (2009)* proposed that the active site $H^+$ wire exits to the protein surface, and leads to Glu69 and Asp72 residues that are located on an electronegative patch on the rim of the active site cavity. On the basis of that observation, these authors proposed that positively charged, protonated buffer molecules dock in that area, from which a proton is delivered to the active site when the enzyme works in the dehydration direction.

Cytosolic CAII has been found to facilitate the transport function of various acid/base transporters including the $Cl^-/HCO_3^-$ exchangers AE1 and AE2 (*Vince and Reithmeier, 1998, 2000*; *Sterling et al., 2001*), the $Na^+/HCO_3^-$ cotransporters NBCe1 and NBCn1 (*Gross et al., 2002*; *Loiselle, 2004*; *Becker and Deitmer, 2007*), and the $Na^+/H^+$ exchanger NHE1 (*Li et al., 2002*, *2006*). Augmentation of acid/base transport by CAII requires both direct binding between transporter and enzyme and CAII catalytic activity, and the complex involved has been coined 'transport metabolon'. The first evidence for a transport metabolon, formed between CAII and an acid/base transporter, was presented in 1993 by *Kifor et al. (1993)* for the $Cl^-/HCO_3^-$ exchanger AE1. CAII could be immunoprecipitated with AE1 when antiserum against the N-terminal of AE1 was used, whereas serum directed against the C-terminal of the transporter failed to immunoprecipitate CAII, suggesting that CAII physically binds to the C-terminal tail of AE1 (*Vince and Reithmeier, 1998*). These findings were confirmed by affinity blotting and by a solid-phase binding assay with CAII and a glutathione S-transferase (GST) fusion protein of the AE1 C-terminal (*Vince and Reithmeier, 1998*). Single-site mutations identified the acidic cluster $D^{887}ADD$ in the C-terminal tail of AE1 as the binding site for CAII (*Vince and Reithmeier, 2000*; *Vince et al., 2000*). Binding of CAII to this cluster would tether the enzyme close to the transporter pore of AE1 near the inner cell surface. This location has been suggested to position CAII ideally to hydrate incoming $CO_2$ and to supply the AE1 transporter directly with a localized substrate pool (*Vince and Reithmeier, 2000*). Indeed, inhibition of CAII catalytic activity decreased the transport activity of AE1, which is heterologously expressed in HEK293 cells, by up to 60% (*Sterling et al., 2001*).

First evidence for a direct interaction between the $Na^+/HCO_3^-$ cotransporter NBCe1 and CAII was presented by *Gross et al. (2002)*, based on isothermal titration calorimetry. In analogy to the CAII-binding cluster $D^{887}ADD$ found in AE1 (*Vince and Reithmeier, 2000*), the cluster $D^{986}NDD$ within the C-terminal of NBCe1 was suggested as the putative CAII binding site. Functional interaction between NBCe1 and CAII was further shown by heterologous protein expression in *Xenopus* oocytes (*Becker and Deitmer, 2007*). Both injection and co-expression of CAII increased NBCe1-mediated membrane current, membrane conductance and $Na^+$ influx when $CO_2$ and $HCO_3^-$ is applied in an ethoxzolamide-sensitive manner.

Evidence for an interaction between NHE1 and intracellular CAII was obtained by measuring the recovery from a $CO_2$-induced acid load in AP1 cells transfected with NHE1 (*Li et al., 2002*). Cotransfection of NHE1 with CAII almost doubled the rate of pH recovery as compared to that in cells expressing NHE1 alone, whereas cotransfection with the catalytically inactive mutant CAII-V143Y even decreased the rate of pH recovery, indicating a physical interaction between NHE1 and catalytically active CAII. Physical interaction between the two proteins was demonstrated by co-immunoprecipitation of heterologously expressed NHE1 and CAII (*Li et al., 2002*). A micro titer plate binding assay with a GST fusion protein of the NHE1 C-terminal tail revealed that CAII binds to the penultimate group of 13 amino acids of the C-terminal tail ($R^{790}$IQRCLSDPGPHP), with the amino acids $S^{796}$ and $D^{797}$ playing an essential role in binding (*Li et al., 2002*, *2006*).

While a considerable amount of data indicates a physical and functional interaction between various acid/base transporters and carbonic anhydrases, several studies have also questioned such transport metabolons. *Lu et al. (2006)* did not observe a CAII-mediated increase in membrane conductance in NBCe1-expressing *Xenopus* oocytes, even when fusing CAII to the C-terminal of NBCe1. In line with these findings, *Yamada et al. (2011)* found no increase in the membrane current during application of $CO_2$ and $HCO_3^-$ when co-expressing wild-type NBCe1A or the mutant NBCe1-Δ65bp (lacking the putative CAII binding site $D^{986}$NDD) with CAII. The concept of a physical interaction between $HCO_3^-$ transporters and CAII has also been challenged by a binding study carried out by *Piermarini et al. (2007)*. These authors were able to reproduce the findings of other groups by showing that sequences in the C-terminal tails of NBCe1, AE1 and NDCBE (SLC4A8) that are fused to GST can bind to immobilized CAII in a micro titer plate binding assay. However, when reversing the assay or using pure peptides, no increased binding of CAII to the immobilized GST fusion proteins could be detected (*Piermarini et al., 2007*). It was concluded that a bicarbonate transport metabolon may exist, but that CAII might not bind directly to the transporters. That CAII activity could improve substrate supply to bicarbonate transporters even without the requirement for a metabolon, or the involvement of direct physical interaction, was also pointed out in a study on AE1 transport activity by *Al-Samir et al. (2013)*. By using Förster resonance energy transfer measurements and immunoprecipitation experiments with tagged proteins, the authors showed no binding or close co-localization of AE1 and CAII. Functional measurements in red blood cells and theoretical modeling suggested that the transport activity of AE1 can be best supported by CAII, when the enzyme is equally distributed within the cell's cytosol (*Al-Samir et al., 2013*). For detailed reviews on transport metabolons see *McMurtrie et al. (2004)*, *Moraes and Reithmeier (2012)*, *Deitmer and Becker (2013)*, *Becker et al. (2014)*.

We have previously shown that transport activity of MCT1 and MCT4 is enhanced by CAII, when the two proteins are heterologously co-expressed in *Xenopus* oocytes (*Becker et al., 2005*, *2010*, *2011*; *Becker and Deitmer, 2008*). In contrast to the transport metabolons described before, enhancement of MCT1/4 transport function is independent of the enzyme's catalytic activity. Both, inhibition of CA catalytic activity with 6-ethoxy-2-benzothiazolesulfonamide (EZA) and co-expression of MCT1/4 with the catalytically inactive mutant CAII-V143Y failed to suppress the CAII-induced facilitation of MCT1/4 transport activity (*Becker et al., 2005*; *Becker and Deitmer, 2008*). The first evidence that this non-catalytic interaction between MCT1/4 and CAII requires direct binding between the two proteins was provided by injection of CAII that was bound to an antibody prior to the injection. In this experiment, CAII was not able to enhance the transport activity of MCT1 in *Xenopus* oocytes, suggesting a sterical suppression of the interaction by the antibody (*Becker and Deitmer, 2008*). In the same study, truncation of the MCT1 C-terminal tail led to loss of the interaction between MCT1 and CAII in *Xenopus* oocytes (*Becker and Deitmer, 2008*). By introduction of single site mutations in MCT1 and MCT4, the glutamic acidic clusters $E^{489}$EE and $E^{431}$EE within the C-terminal tail of MCT1 and MCT4, respectively, could be identified as binding sites for CAII (*Stridh et al., 2012*; *Noor et al., 2015*). In both studies, binding was confirmed both on the functional level by measuring MCT transport activity in *Xenopus* oocytes and by pull-down assays using GST-fusion proteins. Direct interaction between MCT1 and CAII was shown not only in vitro and by heterologous protein expression in *Xenopus* oocytes, but also in an in situ proximity ligation assay in mouse astrocytes (*Stridh et al., 2012*). As CA catalytic activity is not required to facilitate MCT transport function, it was hypothesized that CAII might utilize parts of its intramolecular proton pathway to function as a proton antenna for the transporter (*Almquist et al., 2010*; *Becker et al., 2011*). Protonatable residues with overlapping Coulomb cages could form proton-attractive domains

and could share a proton at a very fast rate, exceeding the upper limit of diffusion-controlled reactions (*Ädelroth and Brzezinski, 2004*; *Friedman et al., 2005*). When these residues are located in proteins or lipid head groups at the plasma membrane, they can collect protons from the solution and direct them to the entrance of a proton-transfer pathway of a membrane-anchored protein, a phenomenon termed a 'proton-collecting antenna' (*Ädelroth and Brzezinski, 2004*; *Brändén et al., 2006*). The need for such a proton antenna is based on the observation that $H^+$ co-transporters, such as MCTs, extract $H^+$ from the surrounding area at rates well above the capacity of simple diffusion to replenish their immediate vicinity. Therefore, the transporter must exchange $H^+$ with protonatable sites at the plasma membrane, which could function as proton collectors for the transporter (*Martínez et al., 2010*). The finding that the mutant CAII-H64A, which is missing the central residue of the enzyme's intramolecular proton shuttle, fails to facilitate MCT transport activity, led to the notion that CAII may function as a proton antenna for the MCT. In doing so, it would provide protons to or or subtract protons from the transporter via its intramolecular $H^+$ shuttle (*Becker et al., 2011*). In line with this, we showed recently that integration of a proton antenna, in the form of six histidine residues, into the C-terminal tail of MCT4 could facilitate MCT4 transport activity even in the absence of carbonic anhydrase (*Noor et al., 2017*).

The transport activity of MCTs was shown to be facilitated not only by intracellular CAII but also by the extracellular CA isoforms CAIV and CAIX (*Klier et al., 2011*, *2014*; *Jamali et al., 2015*). Hypoxia-regulated CAIX is considered to be a key protein in tumor acid/base regulation. CAIX, the expression of which is usually linked to poor prognosis, is tethered to the membrane by a transmembrane domain, with its catalytic center facing the extracellular site. Like other fast carbonic anhydrases, CAIX is equipped with an intramolecular proton shuttle for rapid exchange of $H^+$ between the catalytic center and the surrounding bulk solution. We were able to demonstrate that knockdown of CAIX with short interfering RNA (siRNA) reduced MCT1-mediated lactate transport in hypoxic MCF-7 cells, whereas inhibition of CA catalytic activity with EZA had no effect on MCT1 transport activity (*Jamali et al., 2015*). Co-expression of MCT1/4 with CAIX in *Xenopus* oocytes revealed that CAIX-mediated facilitation of MCT transport activity requires the enzyme's intramolecular proton shuttle, His200 (*Jamali et al., 2015*). Furthermore, knockdown of CAIX, but not inhibition of CA catalytic activity, reduced the proliferation of hypoxic MCF-7 cells, indicating that the CAIX-driven increase in lactate efflux is crucial for the proper functioning of cancer cells (*Jamali et al., 2015*).

In the present study, we investigated whether CAII can facilitate proton-driven lactate transport in MCF-7 breast cancer cells. Knockdown of CAII, which is closely co-localized with MCT1 in MCF-7 cells, resulted in a significant reduction of MCT transport activity in both normoxic and hypoxic cells. We then further analyzed the molecular mechanism underlying the CAII-mediated facilitation of MCT transport activity. Our results showed that CAII features a moiety (Glu69 + Asp72) that exclusively mediates proton exchange with the transporter, while the central residue of its intramolecular proton shuttle (His64) mediates the binding of CAII to MCT1 and MCT4.

## Results

### Intracellular CAII facilitates lactate transport in MCF-7 breast cancer cells

We have recently shown that extracellular CAIX facilitates lactate transport in hypoxic cancer cells by acting as a proton antenna for MCT1/4 at the extracellular face of the plasma membrane (*Jamali et al., 2015*). As proton-driven lactate transport would also benefit from a proton antenna at the cytosolic face of the cell membrane, this work investigated whether proton–lactate co-transport in cancer cells is facilitated by intracellular CAII. We knocked down CAII in normoxic (20% $O_2$) and hypoxic (1% $O_2$) MCF-7 breast cancer cells using siRNA and determined MCT-mediated proton–lactate co-transport by measuring changes in intracellular pH ($pH_i$) during the application and the removal of 3 and 10 mM lactate in the nominal absence of $CO_2$ and $HCO_3^-$ (*Figure 1A,B*). Indeed, knockdown of CAII resulted in a significant decrease in the rate of change in $pH_i$ ($\Delta pH_i/\Delta t$) both during the application and the removal of lactate, indicating a decrease in the lactate transport rate with reduced CAII (*Figure 1C,D*). Interestingly, reduction in transport activity was more pronounced in hypoxic cells than under normoxic conditions (25%–33% reduced in normoxic cells, 47–57%

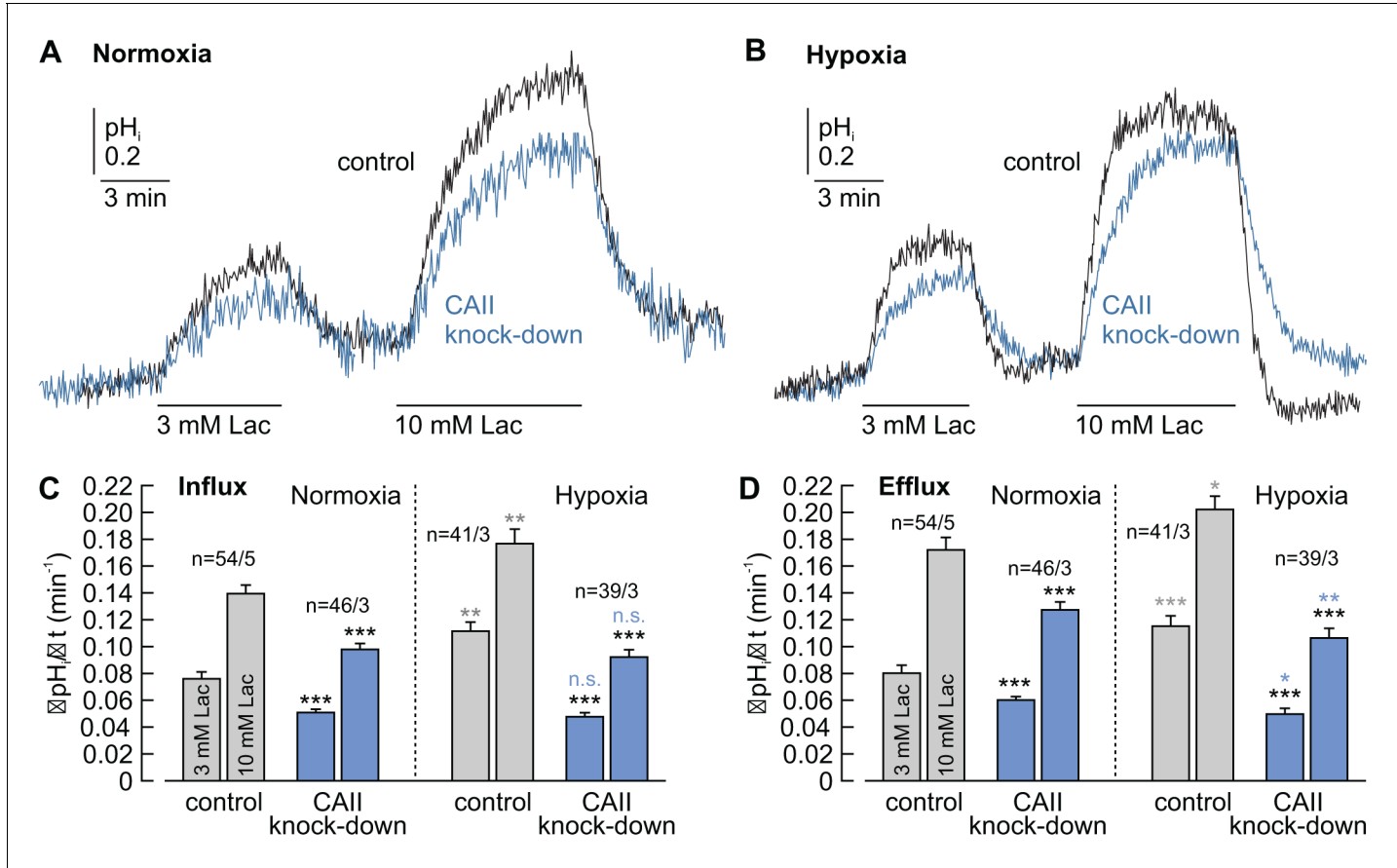

**Figure 1.** CAII facilitates lactate-induced proton flux in MCF-7 breast cancer cells. (**A, B**) Original recordings of lactate-induced changes in intracellular pH (pH$_i$) in normoxic (**A**) and hypoxic (**B**) MCF-7 breast cancer cells treated with either negative control siRNA (control, black traces) or CAII-siRNA (CAII knockdown, blue traces). (**C, D**) Rate of change in intracellular pH ($\Delta$pH$_i$/$\Delta$t) during the application (**C**) and withdrawal (**D**) of lactate in normoxic and hypoxic MCF-7 breast cancer cells treated with either negative control siRNA or CAII-siRNA (mean +SEM). Knockdown of CAII results in a significant reduction of lactate-induced pH change under both normoxic and hypoxic conditions. The black asterisks above the bars for CAII knockdown cells refer to the corresponding bars of the control cells. The blue and gray significance indicators above the bars for hypoxic cells refer to the corresponding bars of normoxic cells. *p≤0.05, **p≤0.01, ***p≤0.001, n.s. no significance; Student's t-test.

DOI: https://doi.org/10.7554/eLife.35176.002

The following source data and figure supplements are available for figure 1:

**Source data 1.** Original dataset for *Figure 1*.
DOI: https://doi.org/10.7554/eLife.35176.006
**Figure supplement 1.** Determination of CAII knockdown efficiency in MCF-7 cells.
DOI: https://doi.org/10.7554/eLife.35176.003
**Figure supplement 2.** Influence of pH$_i$ on lactate transport.
DOI: https://doi.org/10.7554/eLife.35176.004
**Figure supplement 3.** Calibration of SNARF-5 in MCF-7 cells.
DOI: https://doi.org/10.7554/eLife.35176.005

reduced in hypoxic cells), even though the expression of CAII, in contrast to CAIX, is not upregulated under hypoxia (*Figure 1—figure supplement 1*; *Jamali et al., 2015*). A possible reason for this effect might be that the increase in proton–lactate transport, mediated by extracellular CAIX under hypoxia, challenges the need for a faster proton supply at the intracellular site. The knockdown efficiency of CAII was checked by qRT-PCR and by western blot analysis, showing a 60% reduction in normoxic and hypoxic cells (*Figure 1—figure supplement 1*). To investigate whether the CAII-induced augmentation in MCT1 transport activity requires the catalytic activity of the enzyme, lactate transport in MCF-7 cells can be determined in the presence of the CA inhibitor EZA. We used this experiment in a recent study on MCF-7 cells with the same settings as those used in

the present study (*Jamali et al., 2015*). In that study, application of 30 μM EZA had no effect on lactate transport, as measured with the lactate-sensitive FRET nanosensor Laconic in normoxic cells (*San Martín et al., 2013*) and by determining the rate of change in intracellular pH in hypoxic MCF-7 cells. This was true both in the presence and in the absence of 5% $CO_2$ and 15 mM $HCO_3^-$ (see Figures 2,3h,i in *Jamali et al., 2015*). As EZA did not alter lactate transport in MCF-7 cells under any condition, it can be concluded that the observed reduction in lactate transport by knockdown of CAII is independent of the enzyme's catalytic activity.

Knockdown of CAII decreased the basal $pH_i$, as measured at the beginning of the experiment, by approximately 0.1 pH units, both in normoxic and in hypoxic MCF-7 cells (*Figure 1—figure supplement 2A*). This acidification indicates that CAII plays an important role in the acid/base regulation of MCF-7 cells, independent of the oxygen tension, either directly by providing $HCO_3^-$ to the buffer system or indirectly by interacting with the cell's acid/base transporters. As an intracellular acidification leads to a less favorable gradient for $H^+$-coupled lactate influx, the observed reduction in $\Delta pH_i/\Delta t$ during lactate application could also be the result of this change in the $H^+$ gradient. To investigate the influence of $pH_i$ on lactate transport, we plotted $\Delta pH_i/\Delta t$ during the application of 3 mM lactate (which was always carried out as the first pulse) against the initial $pH_i$ (before lactate application) for every individual cell (*Figure 1—figure supplement 2B–E*). No positive correlation between $\Delta pH_i/\Delta t$ and $pH_i$ was observed in all of the four conditions. From these results, it can be concluded that the decrease in $pH_i$, as induced by knockdown of CAII, seems to play only a minor role in the observed reduction in lactate transport.

## CAII is co-localized with MCT1 in cancer cells

We recently investigated which MCT isoforms mediate lactate transport in MCF-7 cells under normoxic and hypoxic conditions (*Jamali et al., 2015*). In that study, application of 300 nM AR-C155858 fully inhibited the lactate-induced acidification of normoxic and of hypoxic MCF-7 cells. AR-C155858 has been shown to inhibit the transport activity of MCT1 (and MCT2 when expressed without its ancillary protein embigin in *Xenopus* oocytes), but has no effect on MCT4 transport activity (*Ovens et al., 2010a, 2010b*). By measuring the rate of change in $pH_i$ during the application of different lactate concentrations, we determined $K_m$ values of ~5 mM lactate from both normoxic and hypoxic MCF-7 cells (*Jamali et al., 2015*). For MCT1, a $K_m$ value of between 3 mM and 8 mM had been determined in various cell types (*Carpenter and Halestrap, 1994*; *Bröer et al., 1998*). For MCT2, the $K_m$ value was found to be 0.74 (*Bröer et al., 1999*; *Heidtmann et al., 2015*), whereas for MCT4, $K_m$ values between 17 mM and 35 mM have been reported (*Dimmer et al., 2000*). Western blot analysis for MCT1, MCT2 and MCT4 revealed sharp bands only for MCT1 in MCF-7 cells under both normoxic and hypoxic conditions (*Jamali et al., 2015*). Taken together, these results indicate that lactate transport in MCF-7 cells is exclusively mediated by MCT1, under both normoxic and hypoxic conditions.

Antibody staining of MCT1 and CAII revealed a homogenous distribution of both proteins within the cells (*Figure 2A*). To investigate whether CAII is colocalized with MCT1 in MCF-7 cells, we performed an in situ proximity ligation assay (PLA). The PLA indicated close proximity (<40 nm) between MCT1 and CAII, in both normoxic and hypoxic cells (*Figure 2B₁, B₂*). The amount of PLA signals per nucleus did not significantly differ between normoxic and hypoxic cells (*Figure 2C*). Knockdown of CAII resulted in a significant reduction in the signal (*Figure 2B₃, C*). When the PLA was carried out without primary antibodies as a negative control, no PLA signals could be detected (*Figure 2B₄, C*).

Taken together, these data suggest that CAII interacts directly with MCT1 to facilitate transport activity.

## CAII supports the proliferation of cancer cells

To investigate whether the CAII-mediated augmentation in lactate flux facilitates cancer cell proliferation, we determined the number of MCF-7 cells kept under different conditions for up to 3 days (*Figure 3A–D*). Knockdown of CAII significantly decreased cell proliferation (as compared to cells transfected with non-targeting negative control siRNA) under both normoxia and hypoxia, whereas transfection with non-targeting negative control siRNA had no significant effects on cell proliferation. Interestingly, total inhibition of lactate transport with 300 nM AR-C155858 decreased cell

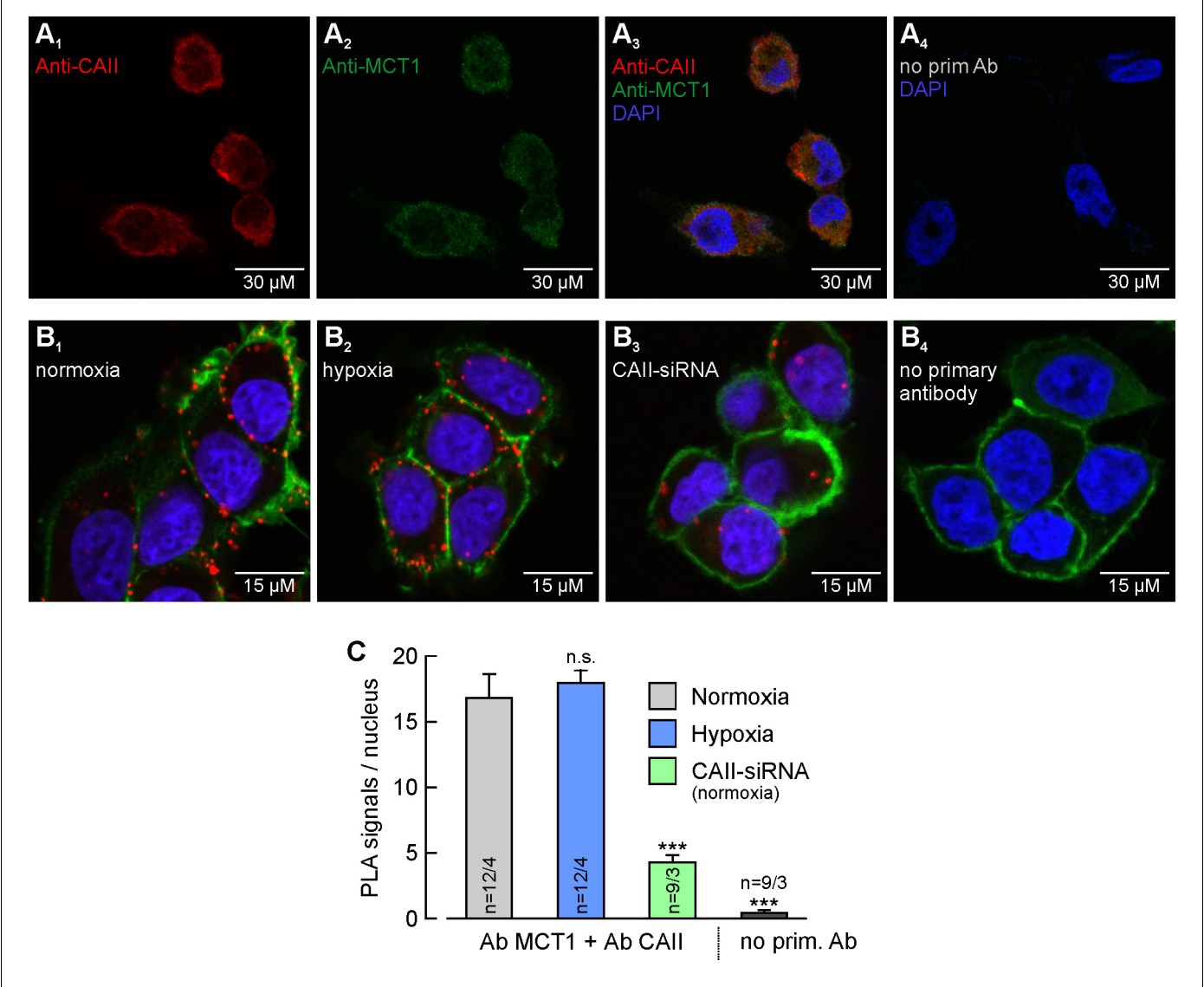

**Figure 2.** MCT1 and CAII are co-localized in MCF-7 breast cancer cells. (**A**) Antibody staining of CAII (**A₁**) and MCT1 (**A₂**) in MCF-7 cells. (**A₃**) Overlay of the fluorescence signals for MCT1 (red), CAII (green) and the nuclei marker DAPI (blue). The specificity of the primary antibodies was tested by incubating MCF-7 cells only with secondary antibodies (**A₄**). (**B**) In situ proximity ligation assay (PLA) of MCT1 and CAII in MCF-7 breast cancer cells, incubated under normoxia (**B₁**) and hypoxia (**B₂**), respectively, and normoxic MCF-7 cells in which CAII was knocked down using siRNA (**B₃**). The red dots indicate co-localization of MCT1 and CAII with a maximum distance of <40 nm. (**B₄**) Negative control of an in situ PLA without primary antibodies. For better visualization of the cells, F-actin was stained with fluorescence-labelled phalloidin (green). (**C**) Quantification of the PLA signals, shown as signals per nucleus (mean +SEM). The significance indicators above the bars refer to the values of the PLA for normoxic cells. ***p≤0.001, n.s. no significance; Student's t-test.

DOI: https://doi.org/10.7554/eLife.35176.007

The following source data is available for figure 2:

**Source data 1.** Original dataset for *Figure 2*.
DOI: https://doi.org/10.7554/eLife.35176.008

proliferation to the same extent as did knockdown of CAII (as compared to untreated cells). Inhibition of CAII catalytic activity with 30 µM EZA, however, had no effect on cell proliferation (as compared to untreated cells). The effects of AR-C155858 and EZA are in line with previous findings that the inhibition of MCT transport activity significantly reduces the proliferation of MCF-7 and MDA-MB-231 cells, whereas inhibition of CA catalytic activity has no effect on cell proliferation

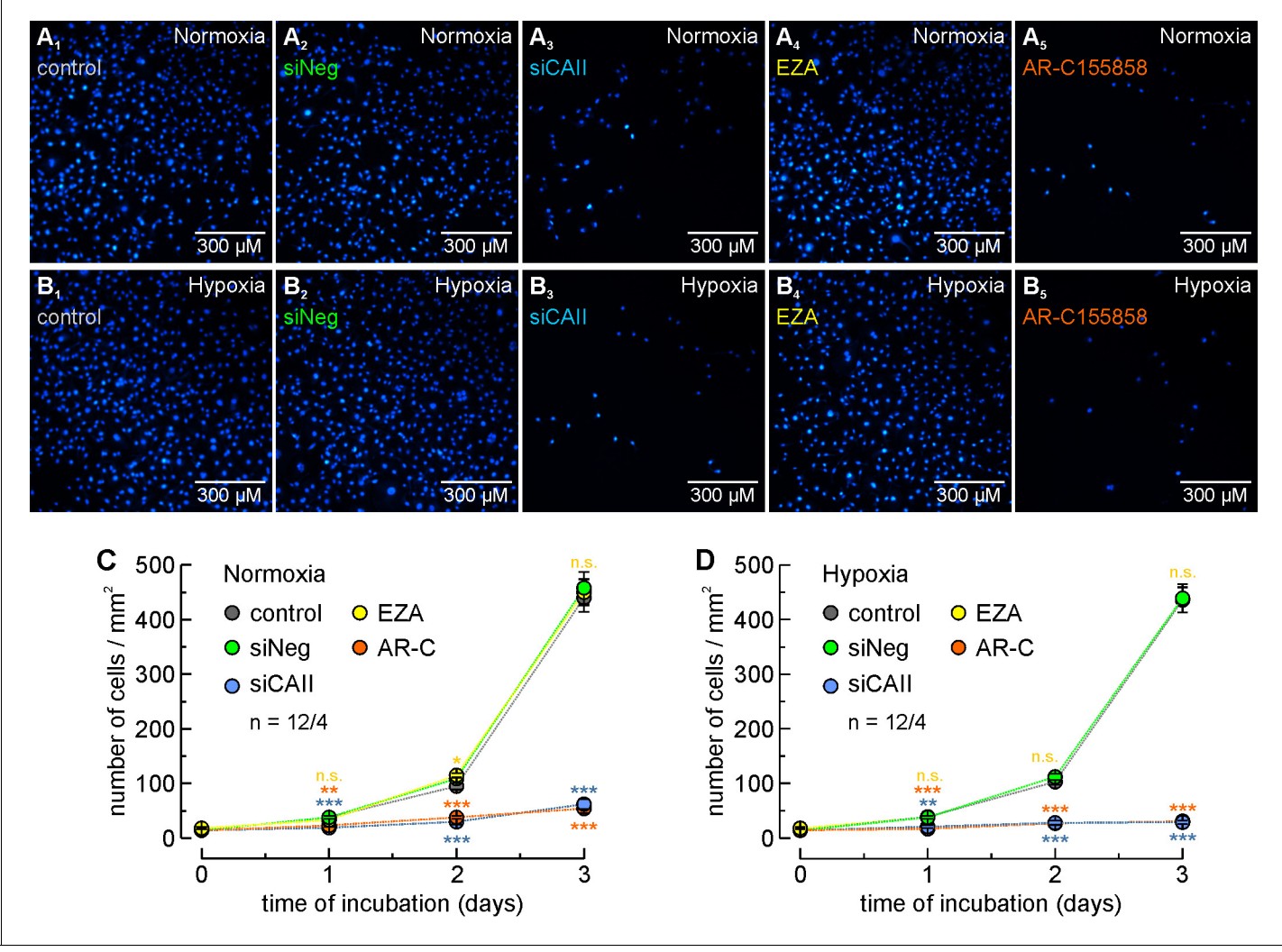

**Figure 3.** CAII supports proliferation of MCF-7 breast cancer cells. (A, B) Staining of nuclei with Hoechst 33342 (blue) in MCF-7 cells after 3 days in culture under normoxic (A) or hypoxic (B) conditions. Cells were untreated (control; A1, B1), mock-transfected with non-targeting negative control siRNA (siNeg; A2, B2), transfected with siRNA against CAII (siCAII; A3, B3), incubated with 30 µM of the CA inhibitor EZA (A4, B4), or incubated with 300 nM of the MCT1 inhibitor AR-C155858 (A5, B5). (C, D) Total number of nuclei/mm$^2$ in normoxic (C) and hypoxic (D) MCF-7 cell cultures, kept for 0–3 days under the conditions described for (A) and (B). For every data point, four dishes of cells were used and three pictures were taken from each dish at random locations, yielding 12 pictures/data points (n = 12/4). The blue asterisks indicate significance in differences between siCAII and siNeg, the orange asterisks between control and AR-C155858, and the yellow significance indicators between control and EZA. *p≤0.05, **p≤0.01, ***p≤0.001, n.s. no significance; Student's t-test.

DOI: https://doi.org/10.7554/eLife.35176.009

The following source data is available for figure 3:

**Source data 1.** Original dataset for *Figure 3*.

DOI: https://doi.org/10.7554/eLife.35176.010

(*Jamali et al., 2015*). The striking effect of CAII knockdown on normoxic and hypoxic cells suggests the possibility that the CAII protein might have yet another role in cell proliferation, in addition to its function as a facilitator of lactate efflux and its catalytic function in cellular pH regulation.

## CAII facilitates a proton-collecting apparatus at its surface to drive MCT transport activity

We have previously shown that CAII facilitates the transport activity of MCT1 and MCT4 when heterologously expressed in *Xenopus* oocytes. CAII-mediated facilitation of MCT1/4 transport activity was

found to be independent of the enzyme's catalytic activity, as both the inhibition of CA catalytic activity with EZA and the co-expression of MCT1/4 with the catalytically inactive mutant CAII-V143Y failed to suppress the CAII-induced facilitation of MCT1/4 transport activity (*Becker et al., 2005*; *Becker and Deitmer, 2008*). As CA catalytic activity is not required to facilitate MCT transport function, it was hypothesized that CAII might utilize parts of its intramolecular proton pathway to function as a proton antenna for the transporter (*Almquist et al., 2010*; *Becker et al., 2011*). To investigate which moieties within the CAII protein could mediate this antennal function, we tested the influence of various CAII mutants on MCT1/4 transport activity using the *Xenopus* oocyte expression system.

On the basis of their work using an algorithm for mapping proton wires in proteins, *Shinobu and Agmon (2009)* suggested that CAII-Glu69 and CAII-Asp72, located on an electronegative patch on the rim of the active site cavity of CAII, could function as a 'docking station' for protonated buffer molecules and thus could facilitate the delivery of protons to the catalytic center. The position of the two amino acids is shown in the cartoon in *Figure 4A*. To test whether CAII-Glu69 or CAII-Asp72 are involved in the facilitation of MCT transport activity, we mutated these two amino acids and the nearby Asp71, and co-expressed the resulting mutants with either MCT1 or MCT4 in *Xenopus* oocytes. The transport activity of MCT1/4 was determined by measuring changes in intracellular proton concentration ($[H^+]_i$) during the application and removal of 3 and 10 mM lactate, while CAII catalytic activity was checked by application of 5% $CO_2$ and 10 mM $HCO_3^-$ (*Figure 4B*; *Figure 4—figure supplement 1A*). CAII-WT increased the transport activity of MCT1 by 80–100%, as measured by the rate of change in $[H^+]_i$ ($\Delta[H^+]_i/\Delta t$) during the application and removal of lactate (*Figure 4C,D*). However, no significant increase in transport activity was observed when MCT1 was co-expressed with CAII-E69Q or CAII-D72N, whereas mutation of the nearby Asp71 (CAII-D71N) had no effect on MCT1 transport activity (*Figure 4B–D*). Co-expression of the CAII mutants with MCT4 gave results similar to those obtained with MCT1 (*Figure 4—figure supplement 1A–C*). Western blot analysis showed that protein expression levels were not altered by the mutations in CAII (*Figure 4—figure supplement 2*).

It has been shown previously that the addition of exogenous $H^+$ donors/acceptors, such as imidazole or carnosine, can rescue proton shuttling in carbonic anhydrase in vitro (*An et al., 2002*; *Tu et al., 1989*). To check whether chemical rescue of the $H^+$ shuttle in CAII-E69Q and CAII-D72N can also restore the functional interaction between these mutants and MCT1, we injected 4-methylimidazole (4-MI) into oocytes co-expressing MCT1 and CAII-E69Q/CAII-D72N. Oocytes expressing MCT1 alone or co-expressing MCT1 +CAII-WT or MCT1 +CAII-D71N were used as control. The effective free volume of an oocyte is around 0.35 µl (*Zeuthen et al., 2002*). Therefore, the injection of 27.6 nl of a 400 mM solution of 4-MI should result in an intracellular 4-MI concentration of 30 mM. Crystallization studies have shown that 4-MI binds to CAII near the His64 by π-stacking to Trp5 and to a moiety near Glu69 and Asp72 (*Figure 5A*; *Duda et al., 2001, 2003*). After the injection of 4-MI, MCT1 transport activity was enhanced after co-expressing CAII-E69Q or CAII-D72N to a similar extent as with co-expression of CAII-WT and CAII-D71N (*Figure 5B–D*). As 4-MI acts as mobile $H^+$ buffer, the compound is expected to increase the cytosolic buffer strength ($\beta_i$), which in turn alters changes in $[H^+]_i$. Therefore, we calculated net $H^+$ fluxes ($J_H$) from the rate of change in intracellular pH during the application of 3 and 10 mM lactate and $\beta_i$ (*Figure 5E*; *Figure 5—figure supplement 1A*). Injection of 4-MI induced a significant increase in lactate-induced $J_H$ in oocytes co-expressing MCT1 with CAII-E69Q or D72N, while leaving $J_H$ unaltered in oocytes co-expressing MCT1 with CAII-WT or CAII-D71N, as well as in oocytes expressing MCT1 alone (*Figure 5E*). Again, the same results could be observed with MCT4 (*Figure 5—figure supplements 1* and *2*). These data indicate that the exogenous $H^+$ donor/acceptor 4-MI can rescue the facilitation of MCT1/4 transport activity by CAII-E69Q and CAII-D72N.

## CAII-Glu69 and Asp72 are not involved in binding between CAII and MCTs

We have previously shown that CAII-mediated enhancement of MCT1/4 transport activity requires direct binding of the enzyme to an acidic cluster within the transporter's C-terminal tail (*Stridh et al., 2012*; *Noor et al., 2015*). To check whether the failure of CAII-E69Q and CAII-D72N to enhance MCT transport activity is due to loss of binding between transporter and enzyme, we performed a pull-down assay with GST fusion proteins of the C-terminal tail of MCT1 or MCT4 (GST-

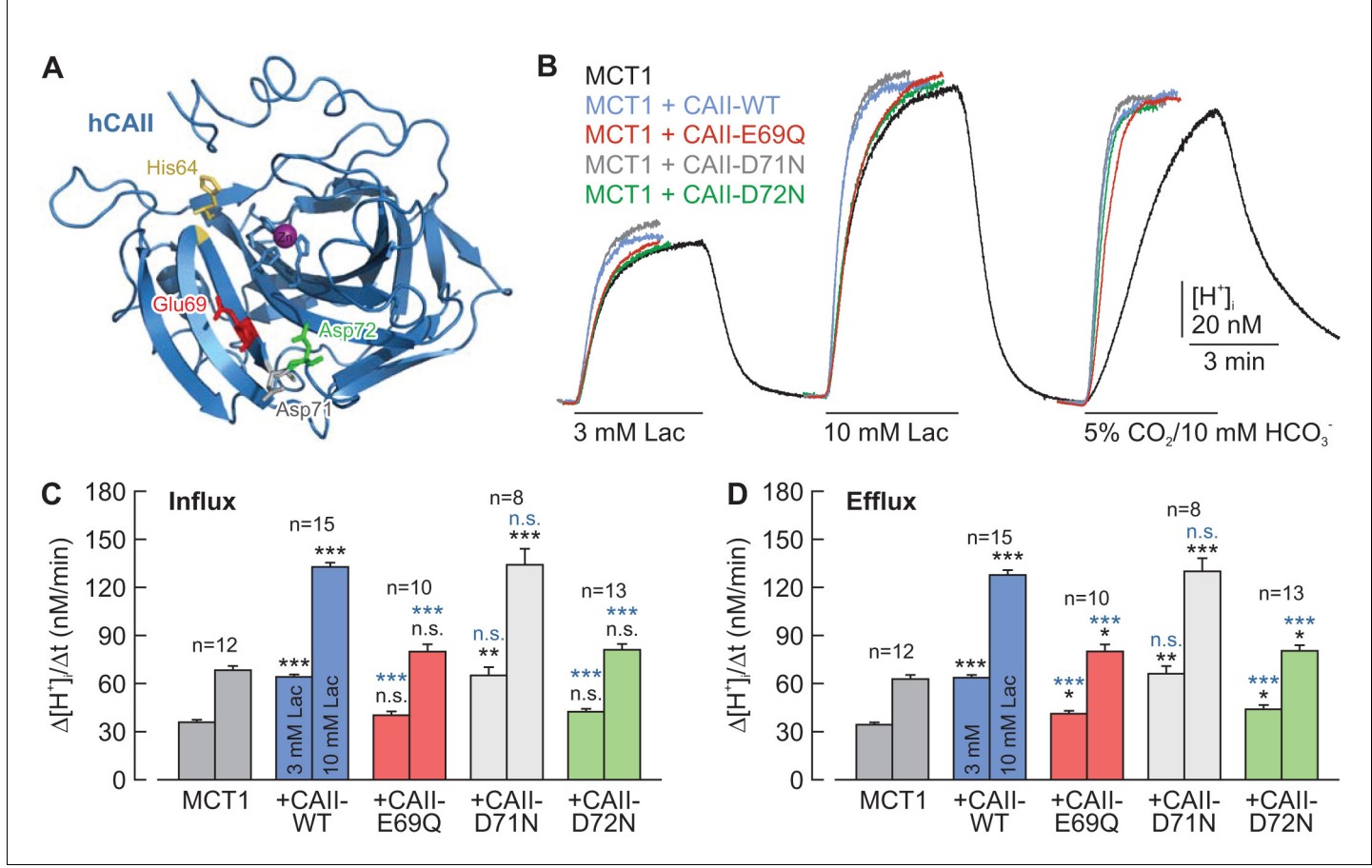

**Figure 4.** CAII-Glu69 and Asp72 are crucial for the facilitation of MCT1 transport activity. (**A**) Structural model of human CAII (PDB-ID: 2CBA; *Håkansson et al., 1992*). Glu69 and Asp72, which have been suggested to form a proton-collecting antenna, are labelled in red and green, respectively. The adjacent Asp71 is labelled in gray. His64, the central residue of the intramolecular proton shuttle, is labelled in yellow. (**B**) Original recordings of the change in intracellular $H^+$ concentration in oocytes expressing MCT1 (black trace), MCT1 +CAII-WT (blue trace), MCT1 +CAII-E69Q (red trace), MCT1 +CAII-D71N (gray trace), and MCT1 +CAII-D72N (green trace), during the application of 3 and 10 mM of lactate and of 5% $CO_2$ and 10 mM $HCO_3^-$. (**C, D**) Rate of change in intracellular $H^+$ concentration ($\Delta[H^+]/\Delta t$) as induced by the application (**C**) and removal (**D**) of 3 and 10 mM lactate, respectively, in oocytes expressing MCT1 (dark gray), MCT1 +CAII-WT (blue), MCT1 +CAII-E69Q (red), MCT1 +CAII-D71N (light gray), and MCT1 +CAII-D72N (green) (mean +SEM). The black significance indicators refer to MCT1, the blue significant indicators refer to MCT1 +CAII-WT. *$p \leq 0.05$, **$p \leq 0.01$, ***$p \leq 0.001$, n.s. no significance; Student's t-test.

DOI: https://doi.org/10.7554/eLife.35176.011

The following source data and figure supplements are available for figure 4:

**Source data 1.** Original dataset for *Figure 4*.
DOI: https://doi.org/10.7554/eLife.35176.014

**Figure supplement 1.** CAII-Glu69 and Asp72 are crucial for the facilitation of MCT4 transport activity.
DOI: https://doi.org/10.7554/eLife.35176.012

**Figure supplement 2.** Expression levels of CAII are not altered by single-site mutation.
DOI: https://doi.org/10.7554/eLife.35176.013

MCT1-CT, GST-MCT4-CT) and lysates of oocytes expressing CAII-WT, CAII-E69Q, CAII-D71N, and CAII-D72N (*Figure 6*). All three CA mutants could be pulled down with GST-MCT1-CT and GST-MCT4-CT, with no evident changes in signal intensity between the mutants and CAII-WT. As negative control, CAII-WT was pulled down with GST alone, resulting in no or only very weak signals. These results indicate that the failure of CAII-E69Q and CAII-D72N to enhance MCT transport activity is not due to a loss of direct binding between transporter and enzyme.

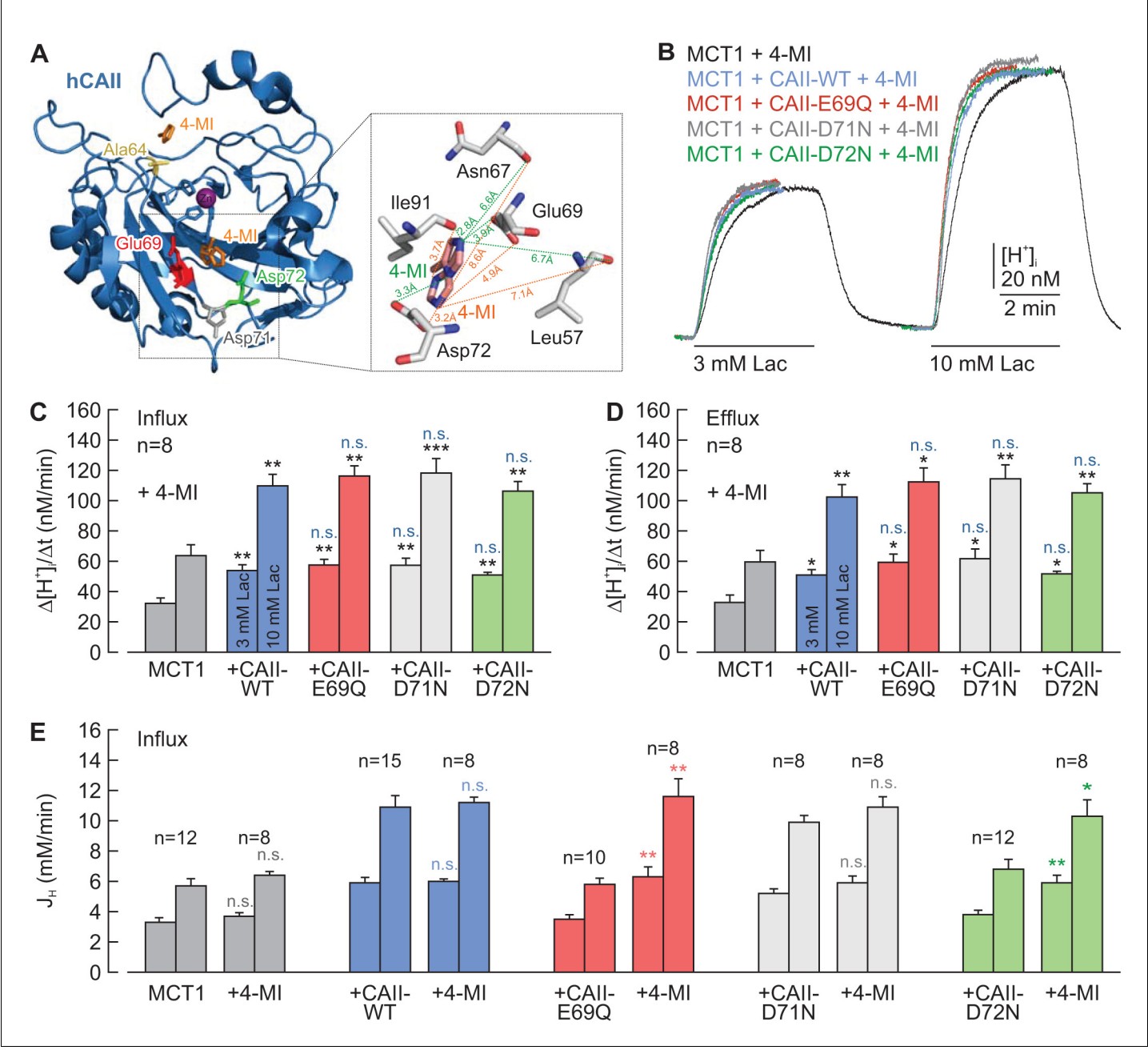

**Figure 5.** Chemical rescue of the interaction between MCT1 and CAII-E69Q/D72N by 4-methylimidazole. (**A**) Structural model of human CAII, complexed with 4-methylimidazole (4-MI) (PDB-ID: 1MOO; *Duda et al., 2003*). 4-MI binds near His64 or in a moiety between Glu69 and Asp 72. Inset: Close-up of the binding moiety. 4-MI can bind in two alternative confirmations (orange and green) between Leu57, Asn67, Glu69, Asp72, and Ile91. (**B**) Original recordings of the change in intracellular $H^+$ concentration in oocytes expressing MCT1 (black trace), MCT1 +CAII-WT (blue trace), MCT1 +CAII-E69Q (red trace), MCT1 +CAII-D71N (gray trace), and MCT1 +CAII-D72N (green trace), during the application of 3 and 10 mM lactate. All oocytes were injected with 4-MI (30 mM) on the day that the measurements were carried out. (**C, D**) Rate of change in intracellular $H^+$ concentration ($\Delta[H^+]/\Delta t$) as induced by the application (**C**) and the removal (**D**) of 3 and 10 mM lactate, respectively, in oocytes expressing MCT1 (dark gray), MCT1 +CAII-WT (blue), MCT1 +CAII-E69Q (red), MCT1 +CAII-D71N (light gray), and MCT1 +CAII-D72N (green) (mean +SEM). The black significance indicators refer to MCT1, the blue significance indicators refer to MCT1 +CAII. All oocytes were injected with 4-MI (30 mM) on the day that the measurements were carried out. (**E**) Lactate-induced proton flux ($J_H$), as calculated from the rate of change in $pH_i$ and the cells intrinsic buffer capacity ($\beta_i$; *Figure 5—figure supplement 1*), in oocytes expressing MCT1 (dark gray), MCT1 +CAII-WT (blue), MCT1 +CAII-E69Q (red), MCT1 +CAII-D71N (light gray), and MCT1 +CAII-D72N (green), either injected with 4-MI or not (mean +SEM). The significance of difference indicators above the bars from 4-MI-injected oocytes refer to comparisons with cells expressing the same proteins without 4-MI. *$p \leq 0.05$, **$p \leq 0.01$, ***$p \leq 0.001$, n.s. no significance; Student's t-test.

*Figure 5 continued on next page*

*Figure 5 continued*

DOI: https://doi.org/10.7554/eLife.35176.015

The following source data and figure supplements are available for figure 5:

**Source data 1.** Original dataset for *Figure 5*.

DOI: https://doi.org/10.7554/eLife.35176.018

**Figure supplement 1.** Intrinsic buffer capacity of oocytes.

DOI: https://doi.org/10.7554/eLife.35176.016

**Figure supplement 2.** Chemical rescue of the interaction between MCT4 and CAII-E69Q/D72N by 4-methylimidazole.

DOI: https://doi.org/10.7554/eLife.35176.017

## CAII-Glu69 and Asp72 do not support CAII catalytic activity

By using an algorithm for mapping proton wires in proteins, *Shinobu and Agmon (2009)* had shown that Glu69 and Asp72 form an extension of the CAII proton wire. From this model, the authors proposed that positively charged, protonated buffer molecules could dock in that area, from which a proton is delivered to the active site to facilitate catalytic activity. To investigate whether Glu69 and Asp72 play a role in CAII catalytic function, we determined CAII catalytic activity in oocyte lysates by gas analysis mass spectrometry (*Figure 7A,B*). CA catalytic activity is determined by the 'log

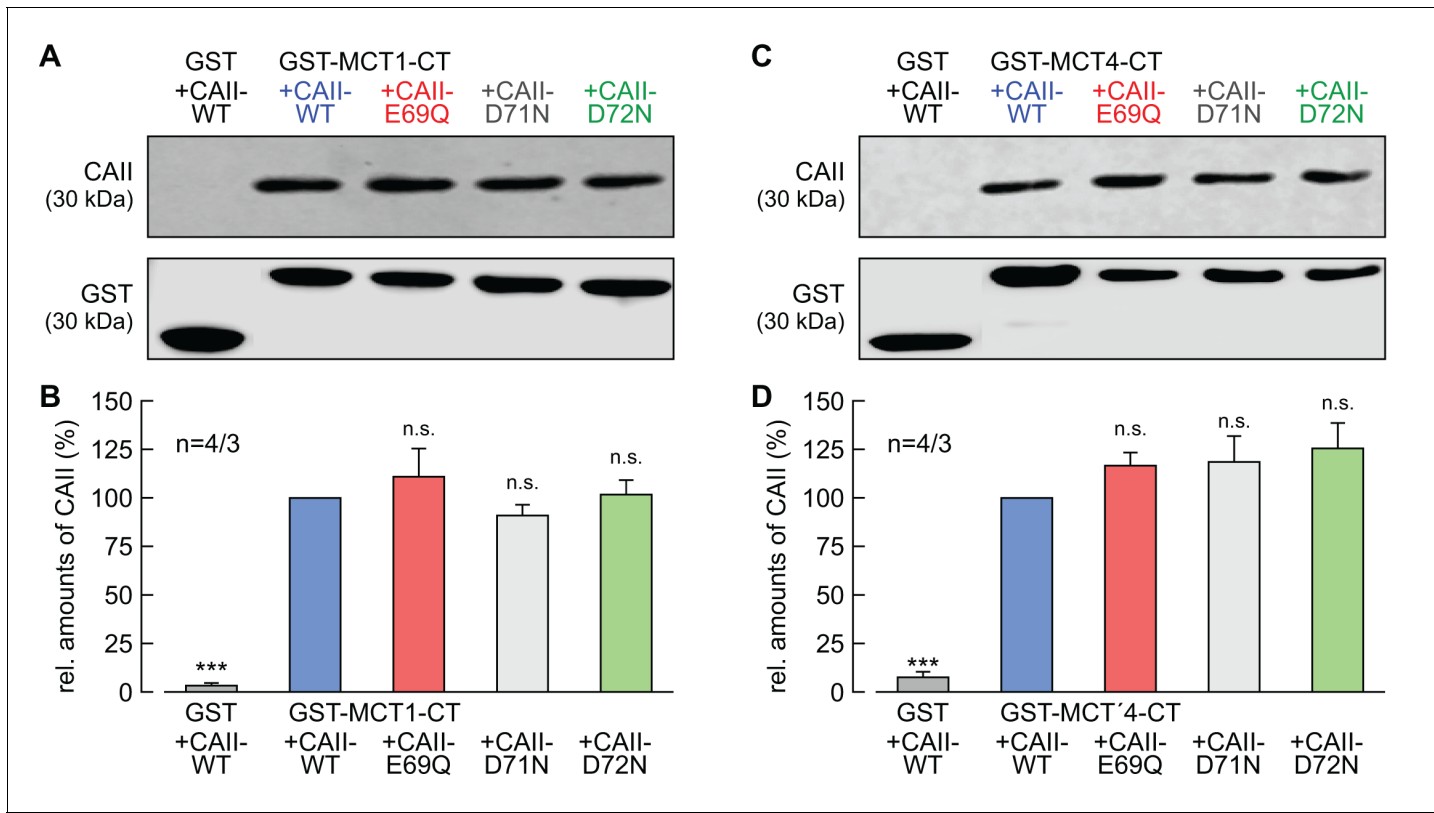

**Figure 6.** CAII-Glu69 and Asp72 do not mediate binding between MCT1/4 and CAII. (A, C) Representative western blots for CAII (upper panel) and GST (lower panel). CAII-WT, CAII-E69Q, CAII-D71N and CAII-D72N were pulled down with a GST fusion protein of the C-terminal of (A) MCT1 (GST-MCT1-CT) or (C) MCT4 (GST-MCT4-CT). As a negative control, CAII-WT was pulled down with GST alone. (B, D) Relative intensity of the fluorescent signal for CAII (mean +SEM). The signal intensity of CAII, pulled down with GST-MCT1/4-CT was set to 100%. The significance indicators refer to the original values of CAII-WT. ***$p \leq 0.001$, n.s. no significance; Student's t-test.

DOI: https://doi.org/10.7554/eLife.35176.019

The following source data is available for figure 6:

**Source data 1.** Original dataset for *Figure 6*.

DOI: https://doi.org/10.7554/eLife.35176.020

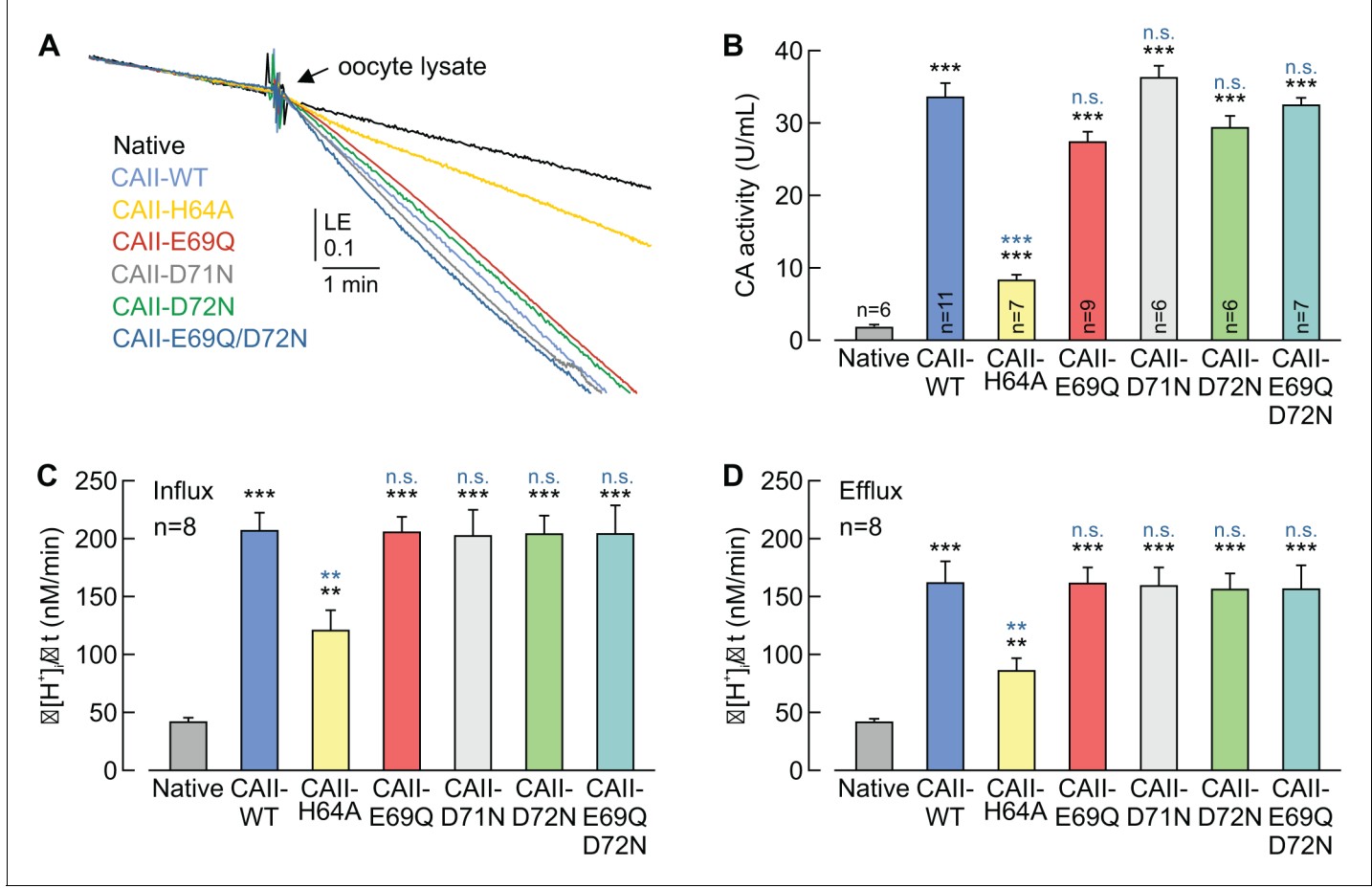

**Figure 7.** CAII-Glu69 and Asp72 do not support CAII catalytic activity. (A) Original recording of the log enrichment (LE) of either a lysate of 20 native oocytes (black trace) or a lysate of 20 oocytes expressing CAII-WT (blue), CAII-H64A (yellow), CAII-E69Q (red), CAII-D71N (gray), CAII-D72N (green), or the double mutant CAII-E69Q/D72N (turquoise). The beginning of the traces shows the rate of degradation of the $^{18}O$-labelled substrate in the non-catalyzed reaction; the black arrow indicates the addition of oocyte lysate. (B) Enzymatic activity of CA in units/ml (mean +SEM). The black asterisks refer to the values from native oocytes, the blue significance indicators refer to the values of oocytes expressing CAII-WT. (C, D) Rate of change in intracellular $H^+$ concentration ($\Delta[H^+]/\Delta t$) as induced by the application (C) and removal (D) of 5% $CO_2$/10 mM $HCO_3^-$ to/from native oocytes and oocytes expressing CAII-WT, CAII-H64A, CAII-E69Q, CAII-D71N, CAII-D72N or the double mutant CAII-E69Q/D72N (mean +SEM). **$p \leq 0.01$, ***$p \leq 0.001$, n.s. no significance; Student's t-test.

DOI: https://doi.org/10.7554/eLife.35176.021

The following source data is available for figure 7:

**Source data 1.** Original dataset for *Figure 7*.

DOI: https://doi.org/10.7554/eLife.35176.022

enrichment' (LE), which is influenced by the rates of both the hydration and dehydration of $CO_2$ (see Materials and methods). Therefore a change in LE indicates changes of CA catalytic activity in both the hydration and the dehydration reaction. Mutation of CAII-Glu69, Asp71, or Asp72 did not alter CAII catalytic activity, whereas mutation of CAII-His64 (the central residue of the CAII intramolecular $H^+$ shuttle) resulted in a 78% reduction in CA activity as compared to that of CAII-WT (*Figure 7A,B*). Even when both Glu69 and Asp72 were mutated together (CAII-E69Q/D72N), no alterations in CAII catalytic activity could be observed (*Figure 7A,B*). The results from the mass spectrometry were confirmed my measuring $\Delta[H^+]_i/\Delta t$ in CAII-expressing oocytes during the application and removal of $CO_2$ and $HCO_3^-$. Expression of CAII resulted in a significant increase in $\Delta[H^+]_i/\Delta t$, as compared to that of native oocytes (*Figure 7C,D*). Mutation of CAII-Glu69, Asp71, Asp72, or double mutation of Glu69/Asp72, resulted in no significant change in CA activity, neither during the application nor during the removal of $CO_2$ and $HCO_3^-$. Only mutation of CAII-His64 induced a significant decrease

in catalytic activity. These data indicate that neither Glu69 nor Asp72 are required for full catalytic activity in CAII. Interestingly, mutation of His64 to Ala reduced CAII catalytic activity to 25%, when measured by gas analysis mass spectrometry (*Figure 7B*), but only to 50%, when measured by $\Delta[H^+]_i/\Delta t$ in intact oocytes (*Figure 7C,D*). These findings infer that the cytosol of oocytes already contains certain, still unidentified, buffer compounds that can support the catalytic activity of the mutant.

## CAII-His64 mediates binding of CAII to MCT1/4, but is not involved in $H^+$ shuttling between the proteins

Our previous studies had shown that CAII-mediated facilitation of MCT transport activity requires CAII-His64, the central residue of the enzymes intramolecular $H^+$ shuttle (*Becker et al., 2011*). Furthermore, our structural models suggest that His64 mediates binding between CAII and an acidic cluster within the C-terminal tails of MCT1 and MCT4 (*Figure 8A*; *Figure 8—figure supplement 1A*; *Stridh et al., 2012*; *Noor et al., 2015*). To investigate whether CAII-His64 mediates binding and/or proton shuttling between MCT1/4 and CAII, we co-expressed MCT1/4 with a CAII in which the histidine at position 64 was mutated either to alanine (CAII-H64A) or to lysine (CAII-H64K). Mutation of His to Ala should disrupt both intramolecular $H^+$ shuttling in CAII and binding of the enzyme to the glutamate residues in MCT1/4. Mutation of His to Lys also disables $H^+$ shuttling, due to the increase in $pK_a$, but should still allow binding of CAII to the glutamate residues in the C-terminal tail of MCT1/4. Indeed, the transport activity of MCT1 and MCT4, as measured by $\Delta[H^+]_i/\Delta t$ during the application and removal of lactate, was not enhanced by co-expression with CAII-H64A, whereas CAII-H64K enhanced MCT transport activity to the same extent as did CAII-WT (*Figure 8B–D*; *Figure 8—figure supplement 1B–D*).

The binding of the CAII mutants to MCT1/4 was evaluated by a pull-down assay with GST fusion proteins of the C-terminal tail of MCT1 or MCT4 (GST-MCT1-CT, GST-MCT4-CT) and using lysates from oocytes expressing CAII-WT, CAII-H64A, or CAII-H64K (*Figure 8E,F*; *Figure 8—figure supplement 1E,F*). CAII-H64A was not able to bind to the C-terminal tail of MCT1 or MCT4, showing only 12% and 14% signal intensity of the band for CAII-WT, respectively, and therefore not being significantly different from the negative control in which CAII-WT was pulled down with GST alone. By contrast, CAII-H64K did still bind to the C-terminal tail of MCT1 and MCT4, with no significant alterations in signal intensity evident in comparisons with the band for CAII-WT.

Catalytic activity of CAII-H64A and CAII-H64K was determined in oocyte lysates by gas analysis mass spectrometry (*Figure 8—figure supplement 2A,B*). Mutations of His64 to either Ala or Lys resulted in a significant decrease in CA catalytic activity with a reduction to 53% (CAII-H64K) and 23% (CAII-H64A), respectively, as compared to CAII-WT. However, mutation of His64 did not change the expression level of CAII in oocytes (*Figure 8—figure supplement 2C,D*). The finding that the catalytic activity of CAII-H64K is strongly reduced compared to CAII-WT, but still significantly higher than the activity of CAII-H64A, indicates that the lysine at position 64 might still be able to function as an intramolecular $H^+$ shuttle, though with significantly less efficacy than His64.

Taken together, these results indicate that His64, the central residue of the CAII intramolecular $H^+$ shuttle, mediates binding of CAII to the transporter's C-terminal tail rather than $H^+$-transfer between the two proteins. However, since the H64K mutation seems to retain some shuttling activity, a role of His64 in $H^+$-transfer cannot be ruled out fully.

## The histidines in the N-terminus of CAII are not involved in binding of CAII to MCT1

It was previously proposed by *Vince et al. (2000)* that CAII binds to the C-terminal tail of the $Cl^-/HCO_3^-$ anion exchanger AE1 via a cluster of five histidine and one lysine residues in the enzyme's N-terminal domain. This assumption was based on the finding that a CAII-mutant (CAII-H3P/H4Q/K9A/H10K/H15K/H17S; coined 'CAII-HEX') that mimics the N-terminal domain of CAI (which does not bind to AE1) failed to bind to the transporter. Indeed the same CAII-mutant was also unable to facilitate MCT1 transport activity (*Becker and Deitmer, 2008*). To investigate the role of His3, His4, Lys9, His10, and His15 in the facilitation of MCT transport activity, we mutated all of these amino acids, either alone or in combination, and co-expressed the resulting mutants with MCT1 in *Xenopus* oocytes. However, none of the mutations resulted in a loss of functional interaction between MCT1

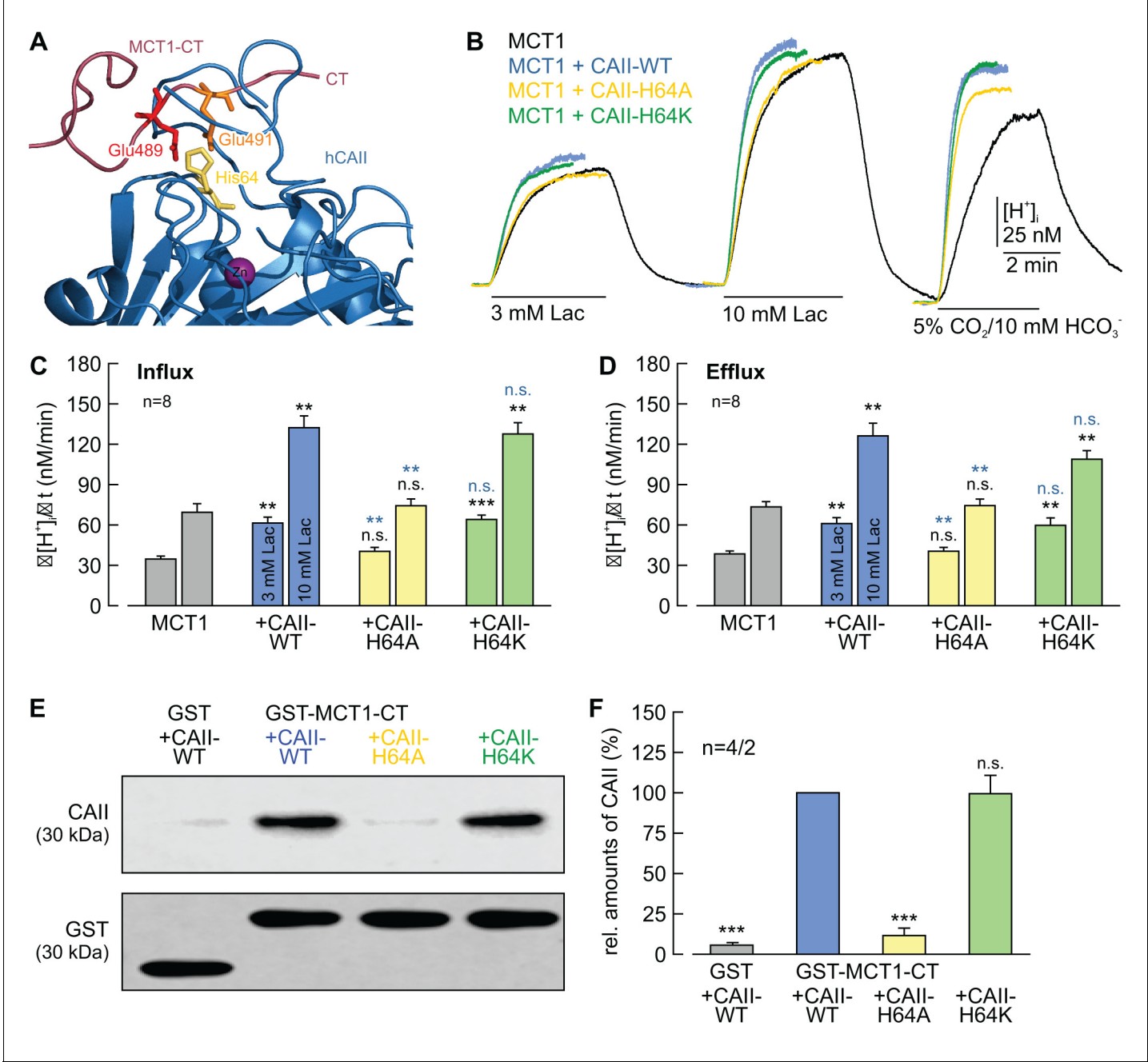

**Figure 8.** CAII-His64 mediates binding, but no proton transfer, between MCT1 and CAII. (**A**) Structural model of the binding between the C-terminal tail of MCT1 (purple) and CAII (blue). Binding is mediated by Glu489 (red) and Glu491 (orange) in the MCT1 C-terminal tail and His64 (yellow) in CAII. (**B**) Original recordings of the change in intracellular $H^+$ concentration in oocytes expressing MCT1 (black trace), MCT1 +CAII-WT (blue trace), MCT1 +CAII-H64A (yellow trace), or MCT1 +CAII-H64K (green trace) during application of 3 and 10 mM of lactate and of 5% $CO_2$ and 10 mM $HCO_3^-$. (**C, D**) Rate of change in intracellular $H^+$ concentration ($\Delta[H^+]/\Delta t$) as induced by the application (**C**) and removal (**D**) of 3 or 10 mM lactate to oocytes expressing MCT1 (gray), MCT1 +CAII-WT (blue), MCT1 +CAII-H64A (yellow), or MCT1 +CAII-H64K (green) (mean +SEM). The black significance indicators refer to MCT1, the blue significance indicators refer to MCT1 +CAII-WT. (**E**) Representative western blots for CAII (upper panel) and GST (lower panel). CAII-WT, CAII-H64A, and CAII-H64K were pulled down with a GST fusion protein of the C-terminal of MCT1 (GST-MCT1-CT). As a negative control, CAII-WT was pulled down with GST alone. (**F**) Relative intensity of the fluorescent signal for CAII (mean +SEM). The signal intensity of CAII pulled down with GST-MCT1-CT was set to 100%. The significance indicators refer to the original values of CAII-WT. ***p≤0.001, n.s. no significance, Student's t-test.

DOI: https://doi.org/10.7554/eLife.35176.023

The following source data and figure supplements are available for figure 8:

*Figure 8 continued on next page*

*Figure 8 continued*

**Source data 1.** Original dataset for *Figure 8*.
DOI: https://doi.org/10.7554/eLife.35176.026
**Figure supplement 1.** CAII-His64 mediates binding, but no proton transfer, between MCT4 and CAII.
DOI: https://doi.org/10.7554/eLife.35176.024
**Figure supplement 2.** Catalytic activity and expression levels of CAII His64 mutants.
DOI: https://doi.org/10.7554/eLife.35176.025

and CAII (*Figure 9*), indicating that none of the histidines or the lysine in the N-terminal of CAII is directly involved in the facilitation of MCT transport activity. A possible reason for the inability of CAII-HEX to facilitate MCT transport activity might be that the introduction of charged and bulky amino acids into the N-terminal domain may prohibit the MCT C-terminal tail from accessing the CAII-His64 binding site.

## Discussion

Our previous studies have shown that CAII facilitates the transport activities of MCT1 and MCT4 independent of the enzymes' catalytic activity, as both inhibition of CA activity with EZA and molecular disruption of the catalytic center (CAII-V143Y) did not reduce the CAII-induced increase in MCT transport activity (*Becker et al., 2005*; *Becker and Deitmer, 2008*). As CA catalytic activity is not required to facilitate MCT transport function, we hypothesized that CAII might utilize parts of its intramolecular proton pathway to function as a proton antenna for the transporter (*Almquist et al., 2010*; *Becker et al., 2011*). The proton pathway of CAII consists of a water wire, coordinated by Asn62 and Asn67, which extends from the active site to the histidine residue at position 64 (*Fisher et al., 2007a, 2007b*). From there, the proton is passed on between the imidazole ring in His64 and exogenous buffer molecules surrounding the enzyme (*Fisher et al., 2007a, 2007b*). In

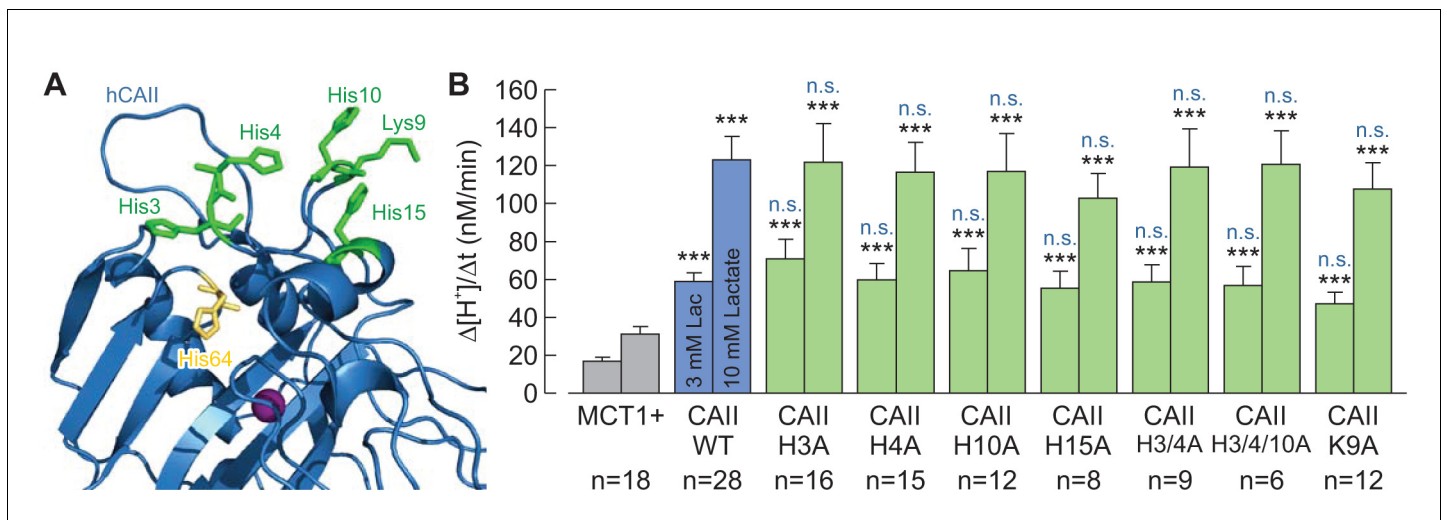

**Figure 9.** The histidine residues in the N-terminus of CAII are not involved in the interaction between CAII and MCT1. (**A**) Structural model of human CAII (PDB-ID: 1XEV). His3, His4, His10, Lys9, and His15, which have been suggested to mediate the binding of CAII to various acid/base transporters, are labelled in green. His64, the binding site for MCT1 and MCT4, is labelled in yellow. (**B**) Rate of change in intracellular $H^+$ concentration ($\Delta[H^+]/\Delta t$), as induced by application of 3 or 10 mM lactate, in oocytes expressing MCT1 (dark gray), MCT1 +CAII-WT (blue), or MCT1 +one of the CAII mutants (green) (mean + SEM). The black significance indicators refer to MCT1, the blue significant indicators refer to MCT1 +CAII-WT. ***$p \leq 0.001$, n.s. no significance; Student's t-test.
DOI: https://doi.org/10.7554/eLife.35176.027

The following source data is available for figure 9:

**Source data 1.** Original dataset for *Figure 9*.
DOI: https://doi.org/10.7554/eLife.35176.028

addition, *Shinobu and Agmon (2009*) presented a model in which the active site $H^+$ wire exits to the protein surface and leads to Glu69 and Asp72, which are located in an electronegative patch on the rim of the active-site cavity. On the basis of this model, the authors proposed that positively charged, protonated buffer molecules could dock in that area, from which a proton is delivered to the active site when the enzyme works in the dehydration direction. However, this assumption is based purely on mathematical modelling and has, to our knowledge, not yet been evaluated experimentally. In the present study, we were able to show that both Glu69 and Asp72 are essential for CAII-mediated facilitation of MCT1/4 transport activity, in both the influx and the efflux direction. Mutation of Glu69 and Asp72 did not, however, alter CAII catalytic activity, as demonstrated in vitro by gas analysis mass spectrometry and in vivo by determining the rate of change in intracellular $H^+$ concentration in oocytes during the application and removal of $CO_2$. These results suggest that this proton-collecting moiety is not involved in the facilitation of CAII catalytic activity, but rather mediates a second, independent function, which is the rapid supply and removal of protons from the pore of the adjacent transporter, rendering CAII a *bona fide* proton antenna. Whether this kind of interaction only exists between CAII and MCTs, or whether CAII could also function as proton antenna for other proton-coupled membrane transporters, such as proton-coupled amino acid transporters or $Na^+/H^+$ exchangers, remains to be shown.

Our previous studies had shown that CAII-mediated facilitation of MCT transport activity requires CAII-His64, which mediates the exchange of protons between the catalytic center and the surrounding bulk solution during the CAII catalytic cycle. Our structural models suggested, however, that His64 mediates binding of CAII to Glu489/Glu491 in MCT1 (*Stridh et al., 2012*) and to Glu431/Glu433 in MCT4 (*Noor et al., 2015*). To investigate whether His64 mediates binding or proton shuttling between MCT and CAII, or both, we exchanged the histidine at position 64 to either alanine or lysine. Mutation of His to Ala should disable both intramolecular $H^+$ shuttling in CAII and binding of the enzyme to the glutamate residues in MCT1/4. Mutation of His to Lys also disables $H^+$ shuttling, as lysine has a considerably higher $pK_a$ than the imidazole ring in histidine, which does not allow fast protonation/deprotonation reactions at physiological pH. However, lysine should still be able to form hydrogen bonds with the glutamate residues in the C-terminal tail of MCT1 and MCT4, and should therefore still enable CAII to bind to the transporter. Although MCT transport activity was not enhanced by CAII-H64A, CAII-H64K facilitated MCT transport activity to the same extent as CAII-WT. These results allowed us to conclude that His64 may not be involved in proton transfer between MCT and CAII, but instead mediates binding of the enzyme to the transporter's C-terminal tail. However, as CAII-H64K still displayed a higher catalytic activity than CAII-H64A, we cannot rule out the possibility that CAII-H64K retains some proton shuttling activity. Indeed, it has been shown that lysine residues that are buried in the protein interior could display considerably lower $pK_a$ values than in those that are exposed to water (*Isom et al., 2011*). Therefore, it might be possible that CAII-H64K can retain enough shuttling activity to facilitate MCT-mediated lactate transport.

Taken together, these results suggest that CAII facilitates MCT transport activity by a mechanism that is completely different from the catalytic activity. Catalytic activity of CAII is facilitated by proton shuttling via His64, whereas Glu69 and Asp72 are ineffective in supporting CA activity. When interacting with MCT1/4, His64 does function as binding site, whereas proton transfer between transporter and enzyme seems to be mediated by Glu69 and Asp72 (*Figure 10*).

The physiological need for CAII's role as a proton antenna for MCTs might derive from the low apparent diffusion rate of protons in a strongly buffered solutionsuch as the cytosol. By applying a mathematical model of Brownian diffusion, *Martínez et al. (2010)* determined the maximum supply capacity of substrates to various transporters and enzymes. For MCT1, they calculated that the diffusion rate of lactate, which is present in the cell at concentrations within the millimolar range, exceeds the turnover rate of the transporter by several magnitudes. This supports the notion that, for this molecule, the cytosol is a well-mixed compartment. For protons, however, *Martínez et al. (2010)* calculated that the maximum supply rate is approximately 15 times lower than the apparent turnover rate of MCT1, as calculated by *Ovens et al. (2010a)*. In other words, the model demonstrates that MCT1 extracts $H^+$ from the cytosol at rates well above the capacity for simple diffusion to replenish its immediate vicinity. This paradoxical result implies that the transporter does not extract $H^+$ substrates directly from the bulk cytosol, but from an intermediate 'harvesting' compartment located between the aqueous phase and the transport site (*Martínez et al., 2010*). The

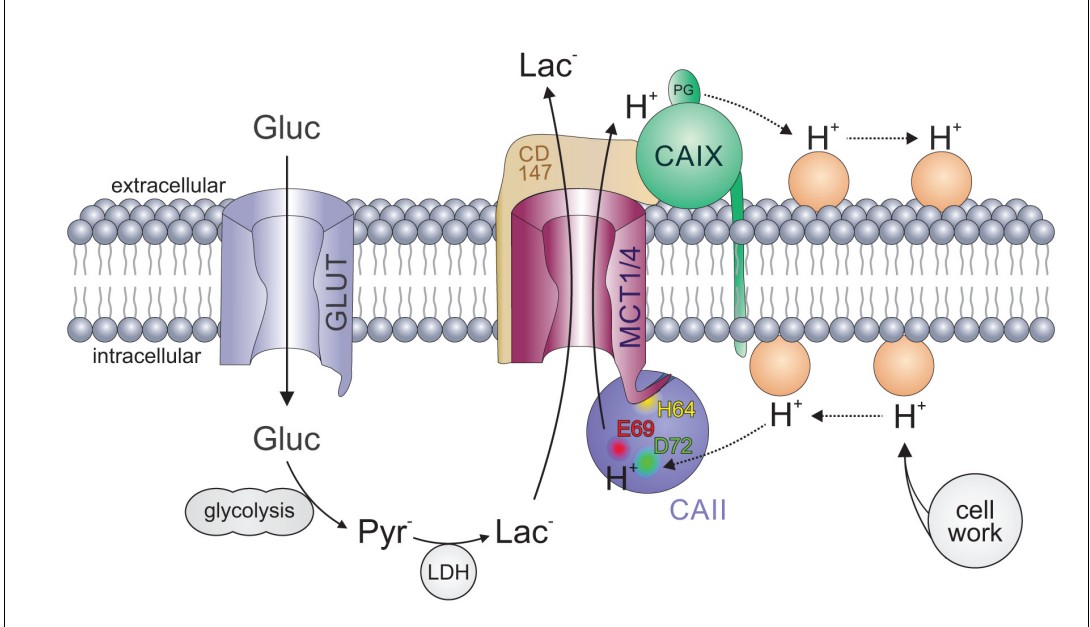

**Figure 10.** Carbonic anhydrases function as proton antennae for MCTs in glycolytic cancer cells. Intracellular CAII (blue circle) is anchored to the C-terminal tail of MCT1/4 (purple structure) via CAII-His64 (yellow spot). This binding brings CAII close enough to the transporter pore to allow the shuttling of protons between the transporter and the surrounding protonatable residues (orange circles). Proton shuttling is mediated by CAII-Glu69 and CAII-Asp72 (red and green dots). Under hypoxic conditions, CAIX (green circle) binds to MCT1 and MCT4 via their chaperon CD147 (ochre structure) to facilitate the exchange of protons between transporter and extracellular protonatable residues (orange circles) via its proteoglycan-like (PG) domain in a fashion similar to that involving Glu69 and Asp72 in CAII. The necessity for this proton antenna derives from the slow diffusion of $H^+$ within the highly buffered cytosol. Lactate, which is produced from glucose (entering the cell by facilitated diffusion via glucose transporters (light blue structure)) by glycolysis and subsequent conversion of pyruvate, quickly reaches the MCT by simple diffusion. Protons, which are produced during the hydrolysis of ATP, by contrast, diffuse very slowly within the cell. To allow fast extrusion of protons and lactate from the cell, the MCT does not extract $H^+$ directly from the bulk cytosol but rather from an intermediate harvesting compartment made up of protonatable residues in the cell membrane and CAII. As in the cytoplasm, the diffusion of ions in the extracellular space is restricted. Therefore protons have to be removed from the extracellular side of the transporter by CAIX and shuttled to protonatable residues on the extracellular face of the cell membrane, from where they can be released to the extracellular space. By this non-catalytic mechanism, intracellular and extracellular carbonic anhydrases could cooperate non-enzymatically to facilitate proton-driven lactate flux across the cell membrane of cancer cells.

DOI: https://doi.org/10.7554/eLife.35176.029

findings of the present study suggest that CAII could serve as such a harvester, which captures protons from protonatable residues at the plasma membrane and funnels them to the transporter pore (*Figure 10*). If this is the case, then CAII would counteract the local depletion of protons in the direct vicinity of the transporter and thus would support transport activity. As the direction of proton movement depends on the ion gradient, this principle can be applied both in the influx and in the efflux direction. In the case of proton/lactate efflux, CAII would harvest $H^+$ from protonatable residues near the plasma membrane and shuttle them to the transporter. In the case of proton/lactate influx, CAII would capture the protons from the transporter pore and distribute them to protonatable residues near the plasma membrane, following the electrochemical $H^+$ gradient. Indeed, we have previously been able to show that protons that are focally applied to MCT1-expressing oocytes moved faster along the inner face of the plasma membrane when CAII was injected, suggesting that CAII could allocate $H^+$ along the plasma membrane and thereby increase the area in which $H^+$ can be released from the membrane into the bulk solution (*Becker and Deitmer, 2008*). In this case, CAII would counteract the accumulation of protons at the transporter pore, which would result in a reduction in transport activity in the influx direction.

Even though the mathematical model from *Martínez et al. (2010)* suggests a 15-fold lower supply rate of $H^+$, our measurements in *Xenopus* oocytes have shown that CAII enhances the transport activity of MCT1 and MCT4 by 'only' around two fold. This implies that, in the absence of CAII, MCT transport activity is either facilitated by other proton-collecting molecules or — more likely — that

MCTs are already able to extract protons from adjacent protonatable sites, even though not as efficiently as when cooperating with CAII.

The relevance of this proton antenna for proper cell function becomes evident from our physiological experiments on MCF-7 breast cancer cells. First, evidence for a functional interaction between MCT and CAII comes from the finding that knockdown of CAII reduces lactate transport in MCF-7 cells. We can rule out the possibility that this decrease in transport activity is due to a loss of CAII catalytic activity as, in a recent study, we were able to show that inhibition of CA activity by application of EZA did not alter lactate flux in MCF-7 cells (*Jamali et al., 2015*). Therefore, it can be assumed that CAII facilitates lactate flux in cancer cells by a non-catalytic, direct interaction. Indeed, CAII is in close co-localization with MCT1 in MCF-7 cells, as shown by an in situ proximity ligation assay. Thus, it appears likely that CAII can form a protein complex with MCT1 in these cells to facilitate lactate flux by functioning as a proton antenna for the transporter. In line with these findings, we previously showed that CAII also facilitates lactate transport in mouse cerebellar and white matter astrocytes, independent of the enzyme's catalytic activity (*Stridh et al., 2012*). As in the cancer cells, MCT1 and CAII are in close proximity in astrocytes, as shown by in situ PLA. Interestingly, colocalization could only be detected when an antibody directed against the intracellular loop between TM 6 and 7 of MCT1 was used. When the PLA was carried out using an antibody against the C-terminal of MCT1, which carries the CAII binding site, no proximity between MCT1 and CAII could be detected (*Stridh et al., 2012*). Astrocytes are highly glycolytic cells, which are widely believed to export lactate as an energy fuel for adjacent neurons, a phenomenon coined an 'Astrocyte-to-Neuron Lactate Shuttle' (*Pellerin and Magistretti, 1994*; *Magistretti, 2006*; *Barros and Deitmer, 2010*). These findings infer that CA-mediated facilitation of MCT1/4 transport activity might be a more general, or at least common, phenomenon in glycolytic cells that have to extrude large amounts of lactate.

In the present study, knockdown of CAII resulted in a more pronounced reduction in MCT1 transport activity in hypoxic than in normoxic cancer cells, but even so, the expression of CAII was not increased under hypoxia. This effect might be attributed to extracellular CAIX, which functions as a proton antenna for MCT1 in MCF-7 cells under hypoxic conditions (*Jamali et al., 2015*). CAIX operates only on the extracellular side of the membrane, which might lead to the formation of a proton micro-domain on the cytosolic side of the membrane in the absence of intracellular CAII because the supply or removal of $H^+$ is enhanced on the extracellular side, leading to an increase in MCT transport activity. This would result in increased accumulation or depletion of $H^+$ at the cytosolic side, depending on transport direction, which would impair MCT transport rate. With a proton antenna on both sides of the membrane, formation of $H^+$ microdomains would be suppressed both at the cis- and at the trans-side. In such a scenario, extracellular CAIX and intracellular CAII would cooperate by a 'push and pull principle', providing protons to the transporter on one side and removing them on the other side of the cell membrane (*Figure 10*). Through this mechanism, intracellular CAII and extracellular CAIX could cooperate to enhance lactate transport in cancer cells under hypoxic conditions. As CAIX is already expressed in normoxic MCF-7 cells at a low level (*Jamali et al., 2015*), this push and pull principle could also occur in normoxic cells, though to a lower extent. The lower, yet robust, expression of extracellular CAIX under normoxia could also explain the reduced effect of CAII knockdown on lactate transport in normoxic MCF-7 cells. In line with this notion, we previously found in *Xenopus* oocytes that intracellular CAII can also work in concert with another extracellular carbonic anhydrase, CAIV, to ensure rapid shuttling of protons and lactate across the cell membrane (*Klier et al., 2014*).

In CAII, $H^+$ shuttling between enzyme and transporter is mediated by Glu69 and Asp72. CAIX seems to lack an analogue moiety within its catalytic domain. However, it features a 59-amino-acid long proteoglycan-like (PG) domain that is unique to CAIX among the CA family (*Pastorek and Pastorekova, 2015*). The PG domain of human CAIX contains 18 glutamate and eight aspartate residues that have been suggested to function as an intramolecular proton buffer, which could support CAIX catalytic activity when operating in an acidic environment (*Innocenti et al., 2009*). We were recently able to show that both the truncation of the CAIX PG domain and the application of a PG-binding antibody (M75) decreased CAIX-induced facilitation in *Xenopus* oocytes and breast cancer cells (*Ames et al., 2018*). Those findings led us to the conclusion that the CAIX PG domain could function as an extracellular proton antenna for monocarboxylate transporters within the acidic environment of a solid tumor.

The importance of CAII for cell viability was demonstrated by a cell proliferation assay. Knockdown of CAII reduced the proliferation of MCF-7 cells to the same extent as did full inhibition of lactate transport with AR-C155858, whereas inhibition of CA catalytic activity with EZA had no effect on proliferation. These results correspond to previous findings that knockdown of CAIX or interference with the CAIX PG domain by an antibody, but not inhibition of CA catalytic activity, decreases cell proliferation in hypoxic MCF-7 and MDA-MB-231 cancer cells (*Jamali et al., 2015*; *Ames et al., 2018*). Apparently, removal of the proton antenna on one side of the membrane is sufficient to reduce MCT-mediated proton/lactate efflux from the cell. This in turn could lead to intracellular lactacidosis and impairment of metabolism, which will ultimately result in a decrease in cell proliferation.

Knockdown of CAII decreased, but did not abolish, lactate transport in MCF-7 cells. However, knockdown of CAII decreased cell proliferation to the same extend as did inhibition of MCT1 transport activity with AR-C155858. This striking effect of CAII knockdown on normoxic and hypoxic cells suggests that the CAII protein might have another role in cell proliferation, which is beyond its function as a facilitator of lactate efflux. As cell proliferation is not decreased by the inhibition of CA activity with EZA, the supporting effect of CAII seems to be independent of the enzyme's catalytic activity. *Zhou et al. (2015)* showed that knockdown of CAII in NCI-H727 and A549 cancer cell lines resulted in significant reduction in clonogenicity in vitro, and in marked suppression of tumor growth in vivo. By using an apoptosis gene array, *Zhou et al. (2015)* found a novel association of CAII-mediated apoptosis with specific mitochondrial apoptosis–associated proteins. These findings suggest that the CAII protein has multiple functions within the cell, including cellular acid/base regulation, facilitation of lactate transport, and regulation of apoptosis.

In summary, our results show that CAII supports lactate flux in cancer cells, presumably by functioning as a proton antenna for MCTs. Therefore, CAII features a molecular moiety that exclusively mediates proton exchange with the transporter, while parts of its intramolecular proton shuttle, which is pivotal for catalytic activity, mediates binding to the MCT. This enhancement of proton movement not only drives basic lactate flux but is also a prerequisite for further facilitation of lactate transport by CAIX under hypoxic conditions, as increased glycolytic activity requires even higher lactate transport capacity in cancer cells.

# Materials and methods

## Key resources table

| Reagent type (species) or resource | Designation | Source or reference | Identifiers | Additional information |
|---|---|---|---|---|
| Cell line (*Homo sapiens*) | MCF-7 | German Collection of Microorganisms and Cell Cultures (DSMZ) | ACC-115 | |
| Recombinant DNA reagent | pGHJ-CAII | *Becker et al., 2005* PMID: 16174776 | | |
| Recombinant DNA reagent | pGHJ-MCT1 | *Bröer et al., 1998* PMID: 9639576 | | |
| Recombinant DNA reagent | pGHJ-MCT4 | *Dimmer et al., 2000* PMID: 10926847 | | |
| Recombinant DNA reagent | pGEX-MCT1-CT | *Stridh et al., 2012* PMID: 22451434 | | |
| Recombinant DNA reagent | pGEX-MCT4-CT | *Noor et al. (2015)* PMID: 25561737 | | |
| Antibody | Rabbit anti-carbonic anhydrase II polyclonal antibody | Millipore | AB1828 | 1:500 |
| Antibody | Goat anti-MCT1 polyclonal antibody | Santa Cruz Biotechnology | sc-14917 | 1:300 |
| Antibody | Mouse anti-β-Actin monoclonal antibody | Santa Cruz Biotechnology | sc-47778 | 1:2500 |
| Antibody | Mouse anti-GST Tag monoclonal antibody | Millipore | 05–782 | 1:400 |

*Continued on next page*

*Continued*

| Reagent type (species) or resource | Designation | Source or reference | Identifiers | Additional information |
|---|---|---|---|---|
| Antibody | Goat anti-rabbit IgG horseradish peroxidase-conjugated secondary antibody | Santa Cruz Biotechnology | sc2004 | 1:2000 |
| Antibody | Goat anti-mouse IgG horseradish peroxidase-conjugated secondary antibody | Santa Cruz Biotechnology | sc-2031 | 1:2000 |
| Antibody | Alexa Fluor 546 donkey anti-rabbit IgG | Invitrogen | A10040 | 1:1000 |
| Antibody | Alexa Fluor 488 donkey anti-goat IgG | Invitrogen | A-11055 | 1:1000 |
| Marker | Alexa Fluor 488 Phalloidin | Life Technologies | A12379 | 1:500 |

## Cultivation of MCF-7 cells

The human breast adenocarcinoma cell line MCF-7 was purchased from the German Collection of Microorganisms and Cell Cultures DSMZ, Braunschweig, Germany (DSMZ-No. ACC-115). Cells were cultured in $35 \times 10$ mm tissue culture dishes (Corning Falcon Easy-Grip Tissue Culture Dish #353001, Fisher Scientific) in RPMI-1640 medium (Sigma-Aldrich, Schnelldorf, Germany), supplemented with 10% fetal calf serum, 2% Minimal Eagle's Medium, 1.7 mM human insulin, 5 mM glucose, 16 mM $NaHCO_3$, and 1% penicillin/streptomycin, pH 7.2, at 37°C in 5% $CO_2$, 95% air (normoxia) or 5% $CO_2$, 1% $O_2$, 94% $N_2$ (hypoxia) in humidified cell culture incubators. Cells were subcultivated for a maximum of 15 passages. The cell line tested negative for contamination with mycoplasma.

## siRNA-mediated knockdown of CAII

CAII was knocked down in MCF-7 cells using siRNA (Ambion Silencer Select anti CA2 siRNA, s2249, Life Technologies). Non-targeting negative control siRNA (Ambion Silencer Select Negative Control No. one siRNA) was used as a control. Cells were transfected with 50 pmol of siRNA, using Lipofectamine RNAiMAX transfection reagent (Life Technologies) in OptiMEM medium (Thermo Fisher). Transfected cells were incubated for 3 days under normoxic or hypoxic conditions. Knockdown efficiency was calculated by measurement of CAII RNA levels with quantitative real-time PCR and of CAII protein levels with western blot analysis.

## Quantitative real-time PCR

Total RNA was extracted from MCF-7 cells, using the RNeasy MinElute Cleanup Kit (Qiagen GmbH, Hilden, Germany). cDNA was synthesized with SuperScript III Reverse Transcriptase (Life Technologies) and random hexamer primers (200 ng; #SO142, Fisher Scientific GmbH, Schwerte, Germany). Real-time PCR was carried out with SYBR Green (PowerUp SYBR Green Master Mix, Applied biosystems) on an AriaMx real-time PCR system (Agiland Technologies). All samples were run in duplicate. RPL27 was used as a reference gene. The characteristics of the primers are shown in *Table 1*. Calculation of the gene expression level was carried out using the comparative threshold method $2^{-\Delta\Delta CT}$.

## Western blot analysis of CAII

To determine expression levels of CAII, MCF-7 cells were harvested by trypsinization and lysed in lysis buffer (50 mM NaCl, 25 mM Tris, 0.5% [v/v] Triton X-100) with protease inhibitor (Roche

**Table 1.** Characteristics of primers used for qRT-PCR

| Gene | Primer | Seq 5'→ 3' | $T_m$ (°C) | Amplicon size (bp) | Location |
|---|---|---|---|---|---|
| CA II | Forward | AAACAAAGGGCAAGAGTGCTGACT | 57.2 | 173 | 680–703 |
| | Reverse | TTTCAACACCTGCTCGCTGCTG | 58.3 | | 831–852 |
| RPL27 | Forward | GGTGGTTGCTGCCGAAATGGG | 58.9 | 101 | 30–50 |
| | Reverse | TGTTCTTCACGATGACAGCTTTGCG | 58.6 | | 106–130 |

DOI: https://doi.org/10.7554/eLife.35176.030

cOmplete, Mini, EDTA-free Protease Inhibitor Cocktail Tablets) for 1 hr at 4°C. The lysate was cleared from the cell debris by centrifugation. Protein concentration was determined with Bradford Reagent (Protein Assay Dye Reagent Concentrate, Bio-Rad). 20 µg of total protein was separated on a 12% polyacrylamide gel and blotted on a polyvinylidene membrane (Roti-PVDF, Carl-Roth). Unspecific binding sites were blocked with 5% milk powder, dissolved in PBS, for 2 hr. CAII was detected using primary anti-CAII antibody (dilution 1:500; rabbit anti-carbonic anhydrase II polyclonal antibody, AB1828, Millipore) and a goat anti-rabbit IgG horseradish peroxidase-conjugated secondary antibody (dilution 1:2000; sc2004, Santa Cruz Biotechnology). After documentation, the membrane was washed 2 × 10 min with stripping buffer (1.5% [w/v] glycine, 0.1% [w/v] SDS, 1% [v/v] Tween20, pH 2.2). As loading control β-actin was labelled with mouse-anti-β-actin (1:2500; sc-47778; Santa Cruz Biotechnology). Quantification of band intensity was carried out with the software ImageJ. The signal for CAII was normalized to the signal for β-actin in the same lane.

## pH imaging in MCF-7 cells

Changes in intracellular pH in MCF-7 cells were measured with the pH-sensitive dye seminaphthorhodafluor 5- and 6-carboxylic acid 5F (Invitrogen SNARF 5F-AM, Life Technologies) using a confocal laser scanning microscope (Zeiss LSM 700 with AxioExaminer.D1 upright microscope) equipped with a 40x water immersion objective (Zeiss C-Apochromat 40x/1.20 W, Carl Zeiss Microscopy GmbH, Frankfurt, Germany). SNARF-5F is a ratiometric pH indicator that exhibits a significant pH-dependent emission shift from yellow-orange to deep red fluorescence when switching from acidic to alkaline pH (*Han and Burgess, 2010*). With a pK$_a$ of 7.2, SNARF-5 is recommended for measuring cytosolic pH$_i$ (*Han and Burgess, 2010*). The fluorescence emission spectrum of SNARF-5 has an isosbestic point at ~590 nm. The isosbestic point is the wavelength at which the fluorescence emission of SNARF-5 does not change under varying pH values, which provides the possibility of ratiometric imaging. For ratiometric imaging, SNARF-5 was excited at 555 nm with a scanning frequency of 0.4 Hz. The emitted light was separated with a variable dichroic mirror at 590 nm in a < 590 nm and a > 590 nm fraction, and the signals of the <590 nm fraction were divided by the signals of the >590 nm fraction.

MCF-7 cells were loaded with 10 µM of the acetoxymethyl ester of SNARF 5 for 10–15 min at room temperature in a 35 × 10 mm tissue culture dish, which was also used as the bath chamber. Loading efficiency was checked by taking snapshots from time to time in the setup. After loading, cells were constantly perfused with medium at room temperature at a rate of 2 ml/min using a tubing system. The medium had the following composition: 143 mM NaCl, 5 mM KCl, 2 mM CaCl$_2$, 1 mM MgSO$_4$, 1 mM Na$_2$HPO$_4$, 10 mM 4-(2-hydroxyethyl)−1-piperazineethanesulfonic acid (HEPES), pH 7.2. In lactate-containing media, NaCl was substituted by Na-L-lactate in equimolar amounts. To measure the rate of lactate-induced acidification reproducibly, cells were depleted of lactate for at least 15 min prior to lactate application. For application of lactate, the application tube was switched between different beakers containing the desired solutions.

Data analysis was carried out with ImageJ (National Institutes of Health, USA). For analysis, a region of interest (ROI) was drawn around each cell (*Figure 1—figure supplement 3 A3*). The intensity of each ROI was recorded for each frame and the values were stored digitally in a spread sheet.

In order to convert the fluorescent ratio into pH values, the system was calibrated by the use of nigericin (10 µM; Life Technologies) as recently described (*Forero-Quintero et al., 2017*). Nigericin is a microbial toxin derived from *Streptomyces hygroscopicus*, which functions as a K$^+$/H$^+$ exchanging ionophore (*Kovbasnjuk et al., 1991*). At high K$^+$ concentrations (130 mM K$^+$ was used in this study), nigericine allows the equilibration of intracellular and extracellular pH. For calibration, SNARF-5 loaded cells were subsequently superfused with saline, and adjusted to pH 6.0, 6.5, 7.0, 7.5, or 8.0 in the presence of nigericin (*Figure 1—figure supplement 3B*). Steady state values were calculated by exponential regression fitting for every single cell (*Figure 1—figure supplement 3B*). The fluorescent ratio was blotted against the extracellular pH and the data were fitted using a Boltzmann function:

$$y = \frac{A_1 - A_2}{1 + e^{\frac{x - x0}{dx}}} + A_2$$

with A1 = initial value, A2 = final value, $x_0$ = center, dx = time constant. The resulting values are given in the inset in *Figure 1—figure supplement 3C*.

On the basis of these data, pH values were calculated from the fluorescent ratio (R) for every data point using the formula

$$pH = \ x_0 + dx * \ln\left(\frac{A_1 - A_2}{R - A_2} - 1\right).$$

The resulting pH values were plotted against the time, as exemplarily shown in *Figure 1A*. From the resulting curve, the maximum rate of change in pH$_i$ ($\Delta pH_i/\Delta t$) during the application of lactate (influx) and the withdrawal of lactate (efflux) was determined for every single cell by linear regression fitting using OriginPro 8.6.

For all imaging experiments, the number of repetitions is given as n = number of cells/number of batches.

## Antibody staining of MCF-7 cells

MCF-7 cells, growing on glass cover slips, were rinsed twice in phosphate buffered saline (PBS) and fixed in 4% paraformaldehyde in PBS for 20 min. Cells were permeabilized with 0.5% Triton X-100 and unspecific binding sites were blocked with 3% bovine serum albumin (BSA) and 1% normal goat serum (NGS) for 2 hr at room temperature. Cells were incubated with primary antibodies (rabbit anti-CAII polyclonal antibody, AB1828, Millipore, dilution 1:500 and goat anti-MCT1 polyclonal antibody, sc-14917, Santa Cruz, dilution 1:300) overnight at 7°C. Cells were washed with PBS and incubated with the secondary antibody (1:1000; Alexa Fluor 546 donkey anti-rabbit IgG and Alexa Fluor 488 donkey anti-goat IgG; Thermo Fisher). Cells were mounted on a microscope slide using mounting medium with DAPI (ProLong Gold Antifade Mountant with DAPI, Thermo Fisher) and analyzed with a confocal laser scanning microscope (Leica TCS SP5, Leica Microsystems, Wetzlar, Germany) with a 63x objective (HCX PL APO lambda blue 63 × 1.4 OIL, Leica Microsystems).

## In situ proximity ligation assay in MCF-7 cells

Colocalization of MCT1 and CAII was examined using the Duolink in situ Proximity Ligation Assay kit (Sigma-Aldrich), as described by the manufacturer and by *Söderberg et al. (2008)*. In short, MCF-7 cells, growing on glass coverslips, were fixed in 4% paraformaldehyde solution (Roti-Histofix 4%; Roth, Karlsruhe, Germany) for 30 min. After fixation, unspecific binding sites were blocked with 3% bovine serum albumin (BSA; Sigma-Aldrich) and 1% normal goat serum (NGS; Sigma-Aldrich) for 2 hr at room temperature in a humid chamber. Cells were incubated with primary antibodies (rabbit anti-CAII polyclonal antibody, AB1828, Millipore, dilution 1:200 and goat anti-MCT1 polyclonal antibody, sc-14917, Santa Cruz, dilution 1:200) for 2 hr at room temperature. The following steps were performed according to the manufacturer's protocol. Briefly, the cells were washed twice and incubated with the PLA probes for one hour at 37°C in a humid chamber. After washing, cells were incubated with ligation-ligase solution. The ligation reaction was carried out at 37°C for 30 min in a humid chamber. After washing, cells were incubated with the amplification-polymerase solution for 100 min at 37°C in a darkened humid chamber. For better visualization of the cells, F-actin was stained with Alexa Fluor 488 Phalloidin (1:500; A12379, Life technologies). Cells were washed again and mounted on a microscope slide using the mounting media supplied with the kit. Cells incubated without primary antibodies were used as procedure controls. The resulting staining was visualized using a confocal laser-scanning microscope (LSM 700; Carl Zeiss GmbH, Oberkochen, Germany), equipped with a Zeiss EC Plan-Neofluar 40x/1.3 objective. Images were analyzed and the PLA signals quantified using ImageJ.

## Measurement of cell proliferation

120 µl of MCF-7 cell suspension ($5 \times 10^5$ cells/ml) was added to 24-well plates, containing 70 µl OptiMEM medium (Thermo Fisher), supplemented either with siRNA (Ambion Silencer Select anti CA2 siRNA, s2249 or Ambion Silencer Select Negative Control No. one siRNA) and lipofectamine RNAiMAX transfection reagent (Life Technologies), or with the CA inhibitor EZA (Sigma Aldrich) and the MCT1 inhibitor AR-C155858 (Santa Cruz Biotechnology). Cells were incubated for 3.5 hr under normoxic conditions in a cell culture incubator. After incubation, the wells were filled up to 0.75 µl with RPMI-1640 medium (Sigma-Aldrich), supplemented with 10% fetal calf serum, 2% Minimal Eagle's Medium, 1.7 mM human insulin, 5 mM glucose, 16 mM NaHCO$_3$, and 1% penicillin/

streptomycin, pH 7.2, and placed at 37°C in 5% $CO_2$, 95% air (normoxia) or 5% $CO_2$, 1% $O_2$, 94% $N_2$ (hypoxia) in humidified cell culture incubators. After the addition of RPMI-1640 medium, the inhibitors were present at a concentration of 30 µM (EZA) and 300 nM (AR-C155858).

Cells were counted after 0 days (immediately after the 3.5 hr incubation period), 1 day, 2 days, and 3 days. Cells were washed with PBS and fixed with 4% paraformaldehyde in PBS for 30 min. After washing, nuclei were stained with 10 µM of the intercalating fluorescent dye Hoechst 33342 (ThermoFisher) for 20 min. Pictures were taken with a fluorescent microscope (Leica DM IRB), equipped with a 10x objective (Leica 10x PH1). Four wells were used for every condition. Three images were taken from each well at random locations, yielding 12 pictures for every condition (n = 12/4). The number of nuclei per image was counted using the program ImageJ. To achieve this, each picture was converted to a monochrome image (with threshold set to 30–255). Fused nuclei were separated with the command 'Watershed' and counted with the command 'Analyze particles' (size set to 0-infinity, circularity set to 0.00–1.00). After counting, the number of nuclei per picture was converted to number of nuclei per $mm^2$.

## Site-directed mutation of CAII

Site-directed mutation of CAII was carried out by PCR using the Phusion High-Fidelity DNA Polymerase (Thermo Fisher) and modified primers that contained the desired mutation. The primers used for creation of the different mutants are shown in *Table 2*. Human CAII, cloned in the oocyte expression vector pGEM-He-Juel, was used as template. PCR was cleaned up using the GeneJET PCR Purification Kit (Thermo Fisher) and the template was digested with DpnI (Fermentas FastDigest DpnI, Thermo Fisher) before transformation into *Escherichia coli* DH5α cells.

## Heterologous protein expression in *Xenopus* oocytes

cDNA coding for human CAII-WT or mutants of CAII, rat MCT1, and rat MCT4, cloned into the oocyte expression vector pGEM-He-Juel, was transcribed in vitro with T7 RNA-Polymerase (mMessage mMachine, Ambion Inc., Austin, USA) as described previously (*Becker et al., 2004*; *Becker, 2014*). *Xenopus laevis* females were purchased from the Radboud University, Nijmegen, The Netherlands. Segments of ovarian lobules were surgically removed under sterile conditions from frogs anaesthetized with 1 g/l of ethyl 3-aminobenzoate methanesulfonate (MS-222, Sigma-Aldrich) and rendered hypothermic. The procedure was approved by the Landesuntersuchungsamt Rheinland-Pfalz, Koblenz (23 177–07/A07-2-003 §6) and the Niedersächsisches Landesamt für Verbraucherschutz und Lebensmittelsicherheit, Oldenburg (33.19-42502-05-17A113). As described

---

**Table 2.** Characteristics of primers used for single-site mutation of CAII
List of the sense primers for single-site mutation CAII. Nucleotides that differ from the wild-type sequence are labelled bold. The antisense primers had the inverse complement sequence of the sense primers. Multiple mutations of CAII were created from single-site mutants using the primers listed in the table. CAII-H64A, which was also used in this study, was kindly provided by Dr. Robert McKenna, University of Florida, Gainesville, U.S.A.

| Mutation | Seq 5'→ 3' | $T_m$ (°C) |
|---|---|---|
| CAII-H3A | CCGAGGATGTCC**GCT**CACTGGGGGTACGGC | 76.3 |
| CAII-H4A | CCGAGGATGTCCCAT**GCC**TGGGGGTACGGC | 76.3 |
| CAII-H3A/H4A | CCGAGGATGTCC**GCTGCC**TGGGGGTACGGC | 77.7 |
| CAII-K9A | GGGTACGGC**GCA**CACAACGGACCTGAG | 72.6 |
| CAII-H10A | GGGTACGGCAAA**GCC**AACGGACCTGAG | 71.0 |
| CAII-H15A | GGACCTGAG**GCC**TGGCATAAGGACTTCC | 71.0 |
| CAII-H64K | CAACAATGGT**AAA**GCTTTCAACG | 57.1 |
| CAII-E69Q | CATGCTTTCAACGTG**CAG**TTTGAT | 59.3 |
| CAII-D71N | GGAGTTT**AAT**GACTCTCAGG | 55.2 |
| CAII-D72N | TTTGAT**AAC**TCTCAGGACAAAGCA | 57.6 |

DOI: https://doi.org/10.7554/eLife.35176.031

previously (*Becker et al., 2004*; *Becker, 2014*), oocytes were singularized by collagenase (Collagenase A, Roche, Mannheim, Germany) treatment in $Ca^{2+}$-free oocyte saline (pH 7.8) at 28°C for 2 hr. The singularized oocytes were left overnight in an incubator at 18°C in $Ca^{2+}$-containing oocyte saline (pH 7.8) to recover. Oocytes of stages V and VI were injected with 5 ng of cRNA coding for MCT1 or MCT4, either together with 12 ng of cRNA coding for CAII or alone. Measurements were carried out 3–6 days after injection of cRNA. 4-MI was dissolved in DEPC-$H_2O$ at a concentration of 400 mM. 27.6 nl of the 4-MI solution was injected on the day of the experiments.

The oocyte saline had the following composition: 82.5 mM NaCl, 2.5 mM KCl, 1 mM $CaCl_2$, 1 mM $MgCl_2$, 1 mM $Na_2HPO_4$, 5 mM HEPES; titrated with NaOH to the desired pH. In $CO_2/HCO_3^-$ and lactate-containing saline, NaCl was substituted by $NaHCO_3$ or Na-L-lactate in equimolar amounts.

## Measurement of intracellular $H^+$ concentration in *Xenopus* oocytes

Changes in intracellular $H^+$ concentration in oocytes were determined with ion-sensitive microelectrodes under voltage-clamp conditions, using single-barreled microelectrodes. For production of the pH-sensitive electrode, a borosilicate glass capillary with filament, 1.5 mm in diameter, was pulled to a micropipette with a tip opening of 0.5–1 µm. To achieve a hydrophobic inner surface, the tip of the micropipette was backfilled with a drop of 5% tri-N-butylchlorosilane in 99.9% pure carbon-tetrachloride using a thin glass capillary and baked for 4.5 min at 450°C on a hot plate under a fume hood. Silanized micropipettes were backfilled with a drop (approximately 0.5 µl) of $H^+$-selective cocktail (Hydrogen ionophore I - cocktail A, 95291, Sigma-Aldrich), using a thin glass capillary, and stored in a wet chamber for at least 30 min so that the viscous cocktail could reach the front end of the tip. Afterwards, the cocktail was covered with 0.1 M Na-citrate solution, pH 6.0, to form a liquid membrane. The electrode was connected to an amplifier (custom build at the University of Kaiserslatern) with a chloride silver wire. For production of the reference and current electrodes, borosilicate glass capillaries with filament, 1.5 mm in diameter, were pulled to micropipettes with a tip opening of 1–2 µm, backfilled with 3 M KCl and connected to an Axoclamp 2B amplifier (Axon Instruments). For calibration of the pH-sensitive electrode, this electrode was superfused with oocyte saline, adjusted to pH 7.0, followed by superfusion with saline adjusted to pH 6.4. Calibration of the electrode was carried out before every single experiment. After calibration of the pH-sensitive electrode, the bath perfusion was stopped and a single oocyte was placed into the bath. The reference and current clamp electrodes were first impaled into the oocyte. Then the pH-sensitive electrode was impaled and positioned onto the 'front' of the oocyte membrane, directly into the direction of flow of the bath perfusion. As the diffusion of protons within the cell is very slow, it is very important to position the pH-sensitive microelectrode as close as possible to the inner face of the plasma membrane, in order to measure fast changes in intracellular $H^+$-concentration (*Bröer et al., 1998*). All experiments were carried out at room temperature (22–25°C). During the whole experiment, the oocyte was superfused with oocyte saline at a flow rate of 2 ml/min using a tubing system. For application of lactate and $CO_2/HCO_3^-$ the application tube was switched between different beakers containing the desired solutions. The measurements were stored digitally using custom-made PC software (*Neumann, 2018*; copy archived at https://github.com/General-Zoology/iClamp and https://github.com/elifesciences-publications/iClamp) based on the program LabView (National Instruments). A step-by-step instruction for the production of ion-sensitive microelectrodes and their use in *Xenopus* oocytes can be found in *Becker (2014)*.

For calculation of pH, the electrode potential ($V_e$) recorded during the calibration was plotted against the pH of the two calibration solutions (pH 7.0 and pH 6.4) and a linear fit was created, using the spread sheet program OriginPro 8.6. Afterwards, the pH values for every recorded data point were calculated using the formula pH = intercept + slope * $V_e$.

Since pH is defined as $-\log([H^+])$, the change in $[H^+]$ at a given change in pH depends on the baseline pH, which could lead to misinterpretation of the real change in proton concentration. For this reason, proton concentration was calculated using the formula:

$[H^+] = 10^{-pH} \times 10^9$ (nM) for every recorded data point.

Transport activity of the MCT was then determined by measuring the rate of change in $[H^+]_i$ ($\Delta[H^+]_i/\Delta t$) during the application or removal of substrate by linear fitting.

To make sure that the change in $[H^+]_i$ during the application of lactate is the result of MCT transport activity and does not depend on the passive diffusion of lactic acid or on endogenous lactate

transporters, we previously determined $\Delta[H^+]_i/\Delta t$ during the application of lactate to *Xenopus* oocytes injected with $H_2O$ instead of cRNA (*Becker et al., 2004*). In $H_2O$-injected oocytes, no lactate-induced changes in $[H^+]_i$ could be observed, indicating that the lactate-induced acidification in MCT-expressing oocytes is the result of MCT-mediated proton-coupled lactate transport (*Becker et al., 2004*).

## Calculation of intrinsic buffer capacity ($\beta_i$) and lactate-induced proton flux ($J_H$)

The intrinsic buffer capacity ($\beta_i$) of oocytes was determined from the pulse of 5% $CO_2$/10 mM $HCO_3^-$, carried out at the end of each experiment (for example see *Figure 4B*). $\beta_i$ was calculated from the change in intracellular pH ($\Delta pH_i$) and the change in intracellular $HCO_3^-$ concentration ($\Delta[HCO_3^-]_i$) during application of $CO_2$ using the formula

$\beta_i = \Delta pH_i / \Delta[HCO_3^-]_i$ (mM).

The exact calculations are shown in*Figure 5—source data 1*.

As all experiments were carried out in the nominal absence of $CO_2$, the $CO_2/HCO_3^-$-dependent buffer capacity ($\beta_{CO_2}$) was omitted from the calculation.

Lactate-induced proton flux ($J_H$) was calculated by multiplying the rate of change in intracellular pH during lactate application ($\Delta pH/t$) with $\beta_i$ using the formula

$J_H = \Delta pH/t \times \beta_i$ (mM).

The calculation of $\beta_i$ and $J_H$ is in *Figure 1—source data 1*.

## Pull-down of CAII

Pull-down of CAII with GST-fusion proteins of the C-termini of MCT1 and MCT4 has been described in detail previously (*Noor et al., 2015*). In brief, the C-termini of MCT1 and MCT4 were cloned into the expression vector pGEX-2T (GE Healthcare Europe GmbH) and transformed into *E. coli* BL21 cells. Protein expression was induced by the addition of 0.8 mM isopropyl-$\beta$-D-thiogalactopyranosid (IPTG). 3 hr after induction, the cells were harvested, resuspended in phosphate-buffered saline (PBS) and lysed in lysis buffer (PBS, supplemented with 2 mM $MgCl_2$, 1% Triton X-100, and protease inhibitor cocktail tablets from Roche). Bacterial lysates were centrifuged for 15 min at 4°C and 12,000 x g, and the supernatant containing the GST-fusion protein (bait protein) was collected for further use. CAII-WT or mutants of CAII were expressed in *Xenopus* oocytes. For each experiment, 25 oocytes were lysed in lysis buffer. Oocyte lysates were centrifuged for 15 min at 4°C and 12,000 x g, and the supernatant (prey protein) was collected for further use.

The pull-down experiment was carried out using the Pierce GST protein interaction pull-down kit (Thermo Fisher). Briefly, for immobilization of GST-fusion protein, bacterial lysate was added to glutathione agarose and incubated for 2 hr at 4°C with end-over-end mixing. fter incubation, the excess bait protein was removed by centrifugation and the beads were washed five times with wash buffer (1 PBS: one lysis buffer). 400 µl of oocyte lysate, containing CAII, was added to the column and incubated for 2 hr at 4°C with end-over-end mixing. After incubation, the excess prey protein was removed by centrifugation and the beads were washed five times with wash buffer. Protein was eluted from the beads with 250 µl of elution buffer (10 mM glutathione in PBS, pH 8.0).

To determine the relative amount of GST and CAII, an equal volume of the samples was analyzed by western blotting. GST was detected using a primary anti-GST antibody (dilution 1:400; anti-GST tag mouse monoclonal IgG, no. 05–782, Millipore) and a goat anti-mouse IgG horseradish peroxidase-conjugated secondary antibody (dilution 1:2000; sc-2031, Santa Cruz). CAII was detected using primary anti-CAII antibody (dilution 1:400; rabbit anti-carbonic anhydrase II polyclonal antibody, AB1828, Millipore) and a goat anti-rabbit IgG horseradish peroxidase-conjugated secondary antibody (dilution 1:2000; Santa Cruz). Quantification of the band intensity was carried out with the software ImageJ. To overcome variations in the signal intensity between different blots, the signal intensity of each band for CAII was normalized to the signal intensity of the band from the pull-down of CAII-WT with the GST-fusion protein of the C-terminal tail. To account for variations in the amount of GST-fusion protein, each normalized signal for CAII was divided by the corresponding normalized signal for GST.

### Determination of CA catalytic activity via mass spectrometry

The catalytic activity of CAII in *Xenopus* oocytes was determined by monitoring the $^{18}O$ depletion of doubly labelled $^{13}C^{18}O_2$ through several hydration and dehydration steps of $CO_2$ and $HCO_3^-$ at 24°C (*Silverman, 1982*; *Sültemeyer et al., 1990*). The reaction sequence of $^{18}O$ loss from $^{13}C^{18}O^{18}O$ (m/z = 49) over the intermediate product $^{13}C^{18}O^{16}O$ (m/z = 47) and the end product $^{13}C^{16}O^{16}O$ (m/z = 45) was monitored with a quadrupole mass spectrometer (OmniStar GSD 320; Pfeiffer Vacuum, Asslar, Germany). The relative $^{18}O$ enrichment was calculated from the measured 45, 47, and 49 abundance as a function of time according to: log enrichment = log (49 × 100/[49 + 47 + 45]). For the calculation of CA activity, the rate of $^{18}O$ degradation was obtained from the linear slope of the log enrichment over the time, using OriginPro 8.6. The rate was compared with the corresponding rate of the non-catalyzed reaction. Enzyme activity in units (U) was calculated from these two values as defined by *Badger and Price (1989)*. From this definition, one unit corresponds to 100% stimulation of the non-catalyzed $^{18}O$ depletion of doubly labelled $^{13}C^{18}O_2$. For each experiment, a batch of 20 native oocytes or 20 oocytes expressing CAII-WT or a mutant of CAIIwas lysed in 80 µl oocyte saline and pipetted into the cuvette. The catalyzed degradation was then determined for 10 min.

### Calculation and statistics

Statistical values are presented as means ± standard error of the mean. Number of repetitions (n) is indicated in each figure panel. For western blot analyses, the number of technical replicates (blots)/ number of biological replicates (cell lysates or pull-down eluates) are separated by a dash. For calculation of significance in differences, Student's t-test was used. In the figures shown, a significance level of $p \leq 0.05$ is marked with *, $p \leq 0.01$ with ** and $p \leq 0.001$ with ***.

## Additional information

### Funding

| Funder | Grant reference number | Author |
|---|---|---|
| Deutsche Forschungsgemeinschaft | BE 4310/6-1 | Holger M Becker |
| Stiftung Rheinland-Pfalz für Innovation | 961-386261/957 | Holger M Becker |
| Research Initiative BioComp | | Joachim W Deitmer Holger M Becker |
| Landesschwerpunkt Membrantransport | | Joachim W Deitmer Holger M Becker |

The funders had no role in study design, data collection and interpretation, or the decision to submit the work for publication.

### Author contributions

Sina Ibne Noor, Conceptualization, Investigation, Visualization, Writing—original draft, Writing—review and editing; Somayeh Jamali, Samantha Ames, Silke Langer, Investigation; Joachim W Deitmer, Conceptualization, Supervision, Funding acquisition, Writing—review and editing; Holger M Becker, Conceptualization, Supervision, Funding acquisition, Visualization, Investigation, Writing—original draft, Writing—review and editing

### Author ORCIDs

Holger M Becker http://orcid.org/0000-0002-2700-6117

### Ethics

Animal experimentation: The procedure of removing oocytes from Xenopus laevis females was approved by the Landesuntersuchungsamt Rheinland-Pfalz, Koblenz (23 177-07/A07-2-003 §6) and

the Niedersächsisches Landesamt für Verbraucherschutz und Lebensmittelsicherheit, Oldenburg (33.19-42502-05-17A113).

## Decision letter and Author response
Decision letter https://doi.org/10.7554/eLife.35176.034
Author response https://doi.org/10.7554/eLife.35176.035

## Additional files

### Supplementary files
• Transparent reporting form
DOI: https://doi.org/10.7554/eLife.35176.032

### Data availability
All data generated or analysed during this study are included in the manuscript and supporting files. Source data files have been provided for Figures 1-9.

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
