## [Decision Letter]

Thank you for submitting your article "A surface proton antenna in carbonic anhydrase II supports lactate transport in cancer cells" for consideration by *eLife*. Your article has been reviewed by three peer reviewers, and the evaluation has been overseen by Matthew Vander Heiden as the Reviewing Editor and Richard Aldrich as the Senior Editor. The following individuals involved in review of your submission have agreed to reveal their identity: Jacques Pouyssegur (Reviewer #1); Robert Gillies (Reviewer #2).

The reviewers have discussed the reviews with one another and the Reviewing Editor has drafted this decision to help you prepare a revised submission.

Summary:

The authors build on a large collection of past work related to the functional interactions between CAII, CAIV, CAIX and the reversible Lactate/H^+^ transporters MCT1 and MCT4. They show that a physical interaction between intracellular CAII and MCT1 or MCT4 can enhance lactate transport across the cell membrane. This effect is independent of CAII catalytic activity, and mutagenesis studies argue that the protons resulting from the catalysis of the CA drive the transport of lactate. This allows them to generalize the concept of 'proton antenna' playing a facilitating role for lactate transport.

Essential revisions:

The reviewers all agreed that the study was well done, and noted that this built on a number of past studies. In light of this, it would be appropriate to more thoroughly discuss how these findings relate to past work and present a coherent model for how the CAs and MCTs function together.

The reviewers also noted that important controls were missing from some experiments, critically verification of CAII knockdown where appropriate.

While it was suggested that extending these studies to additional cell lines might be informative, this was not deemed to be essential. However, the reviewers agreed that assessing how CAII knockdown in MCF7 cells affects cell pH and proliferation would be appropriate.

*Reviewer #1:*

This article completes a large collection of manuscripts published for almost fifteen years by Becker and Deitmer on the functional interactions between CAII, CAIV, CAIX and the reversible transporters Lactate/H^+^ (MCT1 and MCT4). These authors have highlighted a positive cooperation exerted by the protons resulting from the catalysis of the CAs on the drive of the export or importation of lactate. Nevertheless this activation could take place independently of the active site of the CAs. This led them to propose and generalize the concept of 'proton antenna' playing a facilitating role of CAs for Lactic acid transport.

What is the novelty brought here by the authors between the already well explored CAII cytoplasmic form and the MCTs?

Firstly the authors demonstrate within breast cancer cells (MCF7) that CAII is co-localized with MCT1 by the Proximity Ligation Technique, PLA. Secondly they confirm that knockdown of CAII in these cells reduces the transport activity of MCT1. Then the authors focused on studying the molecular mechanisms of CAII and MCT1 interaction and of activation by exploiting Shinobu and Agmon algorithm for mapping H^+^ wires in proteins.

They demonstrated that the His64 residue is not involved in the activation of MCTs by H^+^ shuttling but in the molecular interaction with MCT. Moreover, they establish that the acidic residues Glu69 and Asp72 are neither involved in the interaction with MCT nor in the enzymatic activity of CAII but in contrast they play a key role in the dissipation of H^+^ and the activation of MCTs.

In Conclusion, this work is remarkably well done with exemplary rigor and appropriate controls. These findings provide an insightful molecular and functional mechanistic model of CAs/MCTs cooperation supporting the criticized concept of 'metabolon'.

*Reviewer #2:*

This manuscript investigates the physical interaction between intracellular CAII and Lactate transport via MCTs 1 and 4. The most impressive parts of this work is that they show convincingly that enhancement of lactate transport required physical interactions between CAII and the MCTs and was independent of CAII catalytic activity. This was shown by clever point mutagenesis of the catalytically active H64 to catalytically inactive H64A, which inhibits binding, and H64K, which does not (Figure 7). Proximity ligation assays (Figure 2) showed close apposition of CAII to MCF1 in MCF7 cells.

However, there are two major concerns:

1) The breast cancer story was limited to two figures with missing controls, and was inadequate. To be relevant to cancer, more cell lines could be investigated, along with the controls. Some impact on phenotype would also be required such as, "What impact does CAII knockdown have on resting pH or pH control?" "What impact does stable knockdown or knockout of CAII have on cell or tumor growth?" "Are there breast cancer cells that do not express CAII?"

2) The oocyte work, which technically elegant, appears to be somewhat incremental with respect to other work by this group. They have shown CAII and CAIX interactions with NBC and MCTs in the past using this approach. How does it all fit together? In the absence of other groups working on the CA-MCT-NBC proton wire, it is incumbent for this group, who are the world leaders in this, to synthesize a coherent message and quantitatively discuss the relevance and importance of these proton wires in physiology. For example, "How do all of these MCTs, NBCs, and CAs fit and work together?" "Under what conditions would a deficiency in this behavior affect a cells' fitness and survivability?"

*Reviewer #3:*

Noor et al. build on past work to show that intracellular CAII binding to MCT1 or MCT4 can enhance lactate transport across the cell membrane. They had previously shown that this effect was independent of the catalytic activity of CAII, and similar to what they had shown for extracellular CAIX binding to MCT1/4, they hypothesized that CAII acted as proton "antenna" to facilitate lactate/H^+^ co-transport. This idea, and the residues involved, had also been supported by modeling efforts. Via analysis of various CAII mutants they provide evidence in support of this model, and identify specific residues on CAII that are important for proton transfer but are not involved in the catalytic activity of CAII. They also identify residues important for binding of CAII to MCT, and are not involved in proton shuttling.

Many of the ideas presented are based in prior work, and so there is some question of novelty here for a top journal. With that said, this is a very well done and interesting study and should be of interest.

The following should also be addressed:

A control is needed to confirm knockdown efficiency of CAII with siRNA. CAII expression is referred to in Figure 1—figure supplement 1, but I do not understand the data presentation to evaluate how CAII changes in hypoxia.

---

## [Author Response]

Essential revisions:The reviewers all agreed that the study was well done, and noted that this built on a number of past studies. In light of this, it would be appropriate to more thoroughly discuss how these findings relate to past work and present a coherent model for how the CAs and MCTs function together.

We added more information about the concepts of the proton antenna and the transport metabolon to the Introduction section, where we also added additional information regarding our previous studies: “We have previously shown that transport activity of MCT1 and MCT4 is enhanced by CAII, when the two proteins are heterologously co-expressed in *Xenopus oocytes* (Becker et al., 2005, 2010, 2011; Becker & Deitmer 2008). […] Furthermore, knockdown of CAIX, but not inhibition of CA catalytic activity, reduced proliferation of hypoxic MCF-7 cells, indicating that the CAIX-driven increase in lactate efflux is crucial for proper function of cancer cells (Jamali et al., 2015).”

Furthermore we discussed how the findings in this study relate to our previous findings: “Our previous studies had shown that CAII-mediated facilitation of MCT transport activity requires CAII-His64, which mediates the exchange of protons between catalytic center and surrounding bulk solution during the CAII catalytic cycle. […] Those findings let us to the conclusion that the CAIX PG domain could function as extracellular proton antenna for monocarboxylate transporters within the acidic environment of a solid tumor.”

Finally we altered the model shown in Figure 10 to provide a more general description of how CAs can facilitate lactate flux in cancer cells: “Figure 10. Carbonic anhydrases function as proton antennae for MCTs in glycolytic cancer cells. Intracellular CAII (blue circle) is anchored to the C-terminal tail of MCT1/4 (raspberry structure) via CAII-His64 (yellow spot). […] By this non-catalytic mechanism intracellular and extracellular carbonic anhydrases could cooperate non-enzymatically to facilitate proton-driven lactate flux across the cell membrane of cancer cells.”

The reviewers also noted that important controls were missing from some experiments, critically verification of CAII knockdown where appropriate.

We now checked expression of CAII in MCF-7 cells by western blot analysis. Quantification of the blots showed that application of siRNA decreased expression of CAII by approximately 60% in normoxic and hypoxic cells. These findings are now shown in the new Figure 1—figure supplement 1C, D and are described in the Results subsection “Intracellular CAII facilitates lactate transport in MCF-7 breast cancer cells”.

While it was suggested that extending these studies to additional cell lines might be informative, this was not deemed to be essential. However, the reviewers agreed that assessing how CAII knockdown in MCF7 cells affects cell pH and proliferation would be appropriate.

We calculated the initial pH values and discussed the results in the Results section: “Knockdown of CAII decreased the basal pH_i_, as measured at the beginning of the experiment, by approximately 0.1 pH units, both in normoxic and in hypoxic MCF-7 cells (Figure 1—figure supplement 2A). […] From these results it can be concluded that the decrease in pH_i_, as induced by knockdown of CAII, seems to play only a minor role in the observed reduction in lactate transport.”

We also determined how CAII influences proliferation of MCF-7 cells (new Figure 3). Knockdown of CAII significantly decreased cell proliferation both under normoxia and hypoxia, while transfection with non-targeting negative control siRNA had no significant effects on cell proliferation. Interestingly, total inhibition of lactate transport with 300 nM of AR-C155858 decreased cell proliferation by the same degree as did knockdown of CAII. Inhibition of CAII catalytic activity with 30 µM EZA, however, had no effect on cell proliferation. We describe these findings in the Results subsection “CAII supports proliferation of cancer cells” and in the eighth paragraph of the Discussion.

Reviewer #2:[…] However, there are two major concerns:1) The breast cancer story was limited to two figures with missing controls, and was inadequate. To be relevant to cancer, more cell lines could be investigated, along with the controls. Some impact on phenotype would also be required such as, "What impact does CAII knockdown have on resting pH or pH control?" "What impact does stable knockdown or knockout of CAII have on cell or tumor growth?" "Are there breast cancer cells that do not express CAII?"

We now analyzed the effect of CAII knockdown on resting pH (new Figure 1—figure supplement 2). Discussion of the results can be found in the Results section: “Knockdown of CAII decreased the basal pH_i_, as measured at the beginning of the experiment, by approximately 0.1 pH units, both in normoxic and in hypoxic MCF-7 cells (Figure 1—figure supplement 2A). […] Under none of the four conditions a positive correlation between ΔpH_i_/Δt and pH_i_ could be observed. From these results it can be concluded that the decrease in pH_i_, as induced by knockdown of CAII, seems to play only a minor role in the observed reduction in lactate transport.”

We now determined how CAII influences proliferation of MCF-7 cells (new Figure 3). Indeed, knockdown of CAII significantly decreased cell proliferation both under normoxia and hypoxia, while transfection with non-targeting negative control siRNA had no significant effects on cell proliferation. Interestingly, total inhibition of lactate transport with 300 nM of AR-C155858 decreased cell proliferation by the same degree as did knockdown of CAII. Inhibition of CAII catalytic activity with 30 µM EZA, however, had no effect on cell proliferation. We describe these findings in the Results section: “To investigate whether the CAII-mediated augmentation in lactate flux facilitates cancer cell proliferation, we determined the number of MCF-7 cells kept under different conditions for up to three days (Figure 3A-D). […] The striking effect of CAII knockdown on normoxic and hypoxic cells suggests the possibility that the CAII protein might have yet another role in cell proliferation, in addition to its function as a facilitator of lactate efflux and its catalytic function in cellular pH regulation.”

The findings are further discussed in the Discussion section: “The importance of CAII for cell viability was demonstrated by a cell proliferation assay. Knockdown of CAII reduced proliferation of MCF-7 cells to the same extent as did full inhibition of lactate transport with AR-C155858, while inhibition of CA catalytic activity with EZA had no effect on proliferation. […] These findings suggest that the CAII protein features multiple functions within the cell, including cellular acid/base regulation, facilitation of lactate transport, and regulation of apoptosis.”

To our knowledge there is no study that explicitly demonstrated absence of CAII in a breast cancer cell line. Mallory et al. (2005) showed that CAII is upregulated in MDA-MB-231 cells in the presence of doxorubicin (as mentioned in the second paragraph of the Introduction section).

2) The oocyte work, which technically elegant, appears to be somewhat incremental with respect to other work by this group. They have shown CAII and CAIX interactions with NBC and MCTs in the past using this approach. How does it all fit together? In the absence of other groups working on the CA-MCT-NBC proton wire, it is incumbent for this group, who are the world leaders in this, to synthesize a coherent message and quantitatively discuss the relevance and importance of these proton wires in physiology. For example, "How do all of these MCTs, NBCs, and CAs fit and work together?" "Under what conditions would a deficiency in this behavior affect a cells' fitness and survivability?"

We now added additional information about the concepts of proton wires and transport metabolons in the Introduction section and discussed the present findings more deeply in the light of our previous findings (see Essential revisions).

Reviewer #3:[…] The following should also be addressed:A control is needed to confirm knockdown efficiency of CAII with siRNA. CAII expression is referred to in Figure 1—figure supplement 1, but I do not understand the data presentation to evaluate how CAII changes in hypoxia.

We now checked expression of CAII in MCF-7 cells by western blot analysis. Quantification of the blots showed that application of siRNA decreased expression of CAII by approximately 60% in normoxic and hypoxic cells. Furthermore the data show no significant differences in CAII expression between normoxic and hypoxic cells. These findings are now shown in the new Figure 1—figure supplement 1C, D and are described in the Results subsection “Intracellular CAII facilitates lactate transport in MCF-7 breast cancer cells”.